# STABLE: Shift-Tolerant Allocation via Black–Litterman using Conditional Diffusion Estimates

**Yejun Soun°, Hosung Lee†, Suyoung Park† & U Kang***
Seoul National University, Seoul, South Korea
†DeepTrade Technologies Inc.
mathieu7819@gmail.com°, ukang@snu.ac.kr*

## ABSTRACT

In dynamic financial market characterized by shifting regimes, how can we make effective investment decisions under the changing 1) market regimes and 2) their impact? Among many research fields in financial AI, portfolio allocation stands out as one of the most practically significant areas. Consequently, numerous researchers and financial institutions continually seek approaches that improve the risk–reward trade-off and strive to apply them in real-world investment scenarios. However, achieving robust risk-adjusted performance is extremely challenging, because each asset's return and volatility fluctuate according to the shifting market regime. In response, modern portfolio theory (MPT) addresses this issue by solving for asset weights that maximize a risk–reward objective, using estimates of the return mean and covariance from historical returns. Reinforcement learning (RL) frameworks have been introduced to directly decide portfolio allocations by optimizing risk-adjusted objectives using asset prices and macroeconomic indices. In this work, we propose STABLE (Shift-Tolerant Allocation via Black–Litterman Using Conditional Diffusion Estimates), which combines a diffusion-based generative model that captures regime shifts with an estimation-based portfolio allocation module that maximizes expected risk-adjusted return. STABLE takes macroeconomic context and asset-specific signals as inputs and generates per-stock return trajectories that reflect the prevailing macro regime while preserving firm-specific dynamics. This yields regime-aware predictive return distributions at the single-stock level together with a coherent covariance structure, which are then incorporated as investor views within a Black–Litterman allocation module to obtain risk-diversified portfolio weights. Empirically, STABLE delivers superior portfolio outcomes, achieving up to 122.9% higher Sharpe ratios with reduced drawdowns across major equity markets. It also attains state-of-the-art time-series estimation, lowering MSE by up to 15.7% compared with generative baselines.

## 1 INTRODUCTION

*Given historical macroeconomic data and stock data, how can we make effective investment decisions under changing market regimes?* Because portfolio allocation directly influences investment performance, it is essential not only to aim for high profitability but also to control volatility and drawdowns for practical deployment (Wang et al., 2019; Sun et al., 2022; Niu et al., 2022; Jeon et al., 2024; Lee et al., 2025).

However, to perform robust portfolio optimization under shifting market regimes, we must overcome three key challenges. First, stocks are high-risk assets with substantial exposure to *global macro conditions*, thus failing to jointly model these macro drivers with *firm-specific factors* undermines predictive accuracy for price dynamics. Second, even when both global factors and local factors, it is difficult to know *how strongly* each factor influences each stock, and the influence varies across assets and over time. Third, individual assets in a portfolio are often *correlated* (Yoo & Kang, 2021; Yoo et al., 2021; Soun et al., 2022; Kim et al., 2024), so failing to diversify risk when determining weights can cause severe drawdowns. There have been existing works relying on portfolio

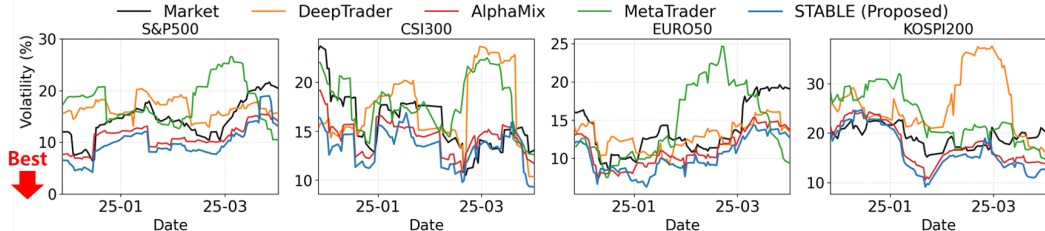

Figure 1: Annualized portfolio volatility across regions. We measure the annualized portfolio volatility in four real-world stock markets (United States, China, Europe, and South Korea). With estimation-driven Black–Litterman optimization, STABLE achieves effective risk diversification and yields the lowest realized volatility across all regions and periods, delivering robust portfolio management.

optimization to maximize risk-adjusted returns (Pun et al., 2020; Sun et al., 2024; Ma et al., 2022), but they depend too much on past asset returns and lack predictive power for future returns, which causes severe performance drops under regime shifts. Others adopt deep reinforcement learning to derive regime-specific strategies (Ye et al., 2020; Hu & Lin, 2019; Gao et al., 2020; Jeon et al., 2026), but they tend to select regimes primarily from macro signals, overfit the prevailing macro state, and fail to capture stock-level idiosyncratic movements.

To overcome the limitations of both classical portfolio theory and deep policy network-based models, we propose STABLE (Shift-Tolerant Allocation via Black–Litterman Using Conditional Diffusion Estimates) that robustly allocates portfolio weights with maximized expected risk-adjusted return. First, STABLE uses a conditional diffusion mechanism to accurately sample per-stock return paths at each step while jointly conditioning on the current macro-regime state and stock-specific features. Second, within the diffusion process, STABLE effectively decomposes per-step noise for each asset into a macro impact and a micro (firm-specific) impact, yielding strong reconstruction performance. Third, STABLE injects the resulting forecasted views into a Black–Litterman allocation module so that the final portfolio weights improve both performance and stability. Fig. 1 shows the cross-region robustness comparison.

Our contributions are summarized as follows:

- **Method.** We present STABLE (Shift-Tolerant Allocation via Black–Litterman Using Conditional Diffusion Estimates), a portfolio allocation method that maximizes expected risk-adjusted return. STABLE couples a conditional diffusion module with Black–Litterman portfolio construction to produce regime-aware predictive distributions for portfolio optimization. STABLE addresses central shortcomings of conventional mean–variance (MVO) approaches and deep RL methods.
- **Exclusive experiments.** We evaluate STABLE on four regional stock markets (United States, China, Europe, and South Korea) with two tasks: portfolio allocation and time-series estimation. STABLE achieves state-of-the-art performance in both tasks. In the portfolio task, STABLE improves the Sharpe ratio by up to 122.9% over the best competitor, demonstrating superior risk-adjusted return. In the estimation task, STABLE improves the MSE metric by up to 15.7% over the best competitor, indicating accurate stock time-series prediction.
- **Case study.** Visual inspection shows that the temporal stock embeddings faithfully reflect real-world sector relationships and allow stocks with similar characteristics at each time step to reference one another. This enables STABLE to condition on intrinsic stock properties, which in turn improves time-series estimation accuracy.

Symbols are summarized in Appendix A.1.

## 2 RELATED WORK

We categorize existing portfolio allocation methods based on how they determine asset weights. Broadly, they can be divided into modern portfolio theory (MPT) methods and deep reinforcement learning approaches.

**Modern portfolio theory.** Classical MPT seeks closed-form solutions for weight allocation by solving a mean–variance optimization (MVO) problem (Markowitz, 1952b). Representative examples include Black–Litterman and robust portfolio optimization. The Black–Litterman model combines investor views with Markowitz-style portfolio optimization to determine allocation (Black & Litterman, 1990), while robust portfolio approaches define an ambiguity set to account for uncertainty in asset returns and then allocate weights based on worst-case scenarios (Goldfarb & Iyengar, 2003). Although these methods strengthen traditional portfolio optimization by *plugging in* estimates of the mean and covariance of stocks, they are effective only when those estimates are accurate at each rebalancing time. However, because their estimation is typically restricted to historical windows, any post-allocation regime shift that drives the realized distribution depart from the past severely degrades both profitability and stability. Instead, STABLE leverages a generative sampling model conditioned on both macro-level and micro-level signals to improve estimation accuracy. The generative sampling process yields per-stock predictive distributions from which the mean and covariance are computed, enabling more effective plug-in to the allocation stage (see Section 3.5).

**Deep reinforcement learning.** Another line of work applies deep reinforcement learning to portfolio allocation. Under this paradigm, a policy network learns to output actions (i.e., portfolio weights) to maximize cumulative rewards such as risk-adjusted return metrics. Early approaches rely on a neural network that directly proposes allocations. Subsequent research incorporates market-regime considerations, aiming to adapt allocations more flexibly. For instance, Alphastock (Wang et al., 2019) introduces asset-axis attention to capture correlations among multiple assets, improving weight computation. Similarly, MetaTrader (Niu et al., 2022) proposes a strategy that selects from typical financial domain baselines (e.g., constant rebalanced portfolio, Markowitz portfolio) under different market regimes. AlphaMix (Sun et al., 2023) employs a routing mechanism that switches between multiple neural network models depending on market conditions, thus incorporating regime awareness into the policy. Despite using regime-aware policy networks, these approaches overlook the aspect that the degree to which a given macro state influences returns is stock-specific. Consequently, their portfolio allocation overfits the macro state and fails to capture stock-level idiosyncratic movements. In contrast, STABLE employs a *learnable guidance scale* to decompose per-step noise into a macro impact and a firm-specific impact, enabling fine-grained per-stock distribution estimation and downstream portfolio allocation (see Sections 3.3 and 3.4).

# 3 PROPOSED METHOD

## 3.1 PROBLEM DEFINITION

**Given** a macro condition $m_\tau \in \mathbb{R}^{d_m}$, per-stock conditions $c_\tau^{(s)} \in \mathbb{R}^{d_c}$, a prior window length $\nu$, a prior mean $\mu_{\text{prior},\tau} \in \mathbb{R}^S$ and prior covariance $\Sigma_{\text{prior},\tau} \in \mathbb{R}^{S \times S}$ computed from the most recent $\nu$ business days, and an investment horizon $\ell$,

**Allocate**

$$w_\tau^\star \in \arg\max_{w_\tau} \frac{\mathbb{E}\big[w_\tau^\top R_{\tau,\tau+\ell}\big]}{\sqrt{\text{Var}(w_\tau^\top R_{\tau,\tau+\ell})}},$$

where $R_{\tau,\tau+\ell} \in \mathbb{R}^S$ denotes the realized adjusted-close returns over the next $\ell$ business days for the $S$ stocks,

**Such that** the budget constraint holds: $\mathbf{1}^\top w_\tau = 1$.

## 3.2 OVERVIEW

To solve the portfolio allocation problem defined in Section 3.1, our proposed STABLE executes three stages as summarized in Figure 2. First, the **Conditional Diffusion Generator (CDG)** generates per-stock return segments conditioned on macro context and firm-specific properties at an individual stock level. Second, **Multi-Level Guidance (MLG)** constructs the estimated noise by combining the shared impact with the idiosyncratic component, modulated by a learnable gate, yielding regime-aware per-stock distributions. Third, a **Black–Litterman–based Mean–Variance Optimizer (BL–MVO)** consumes the diffusion-induced moments as views and outputs allocation weights that balance diversification and estimation. The main challenges and our ideas are as follows.

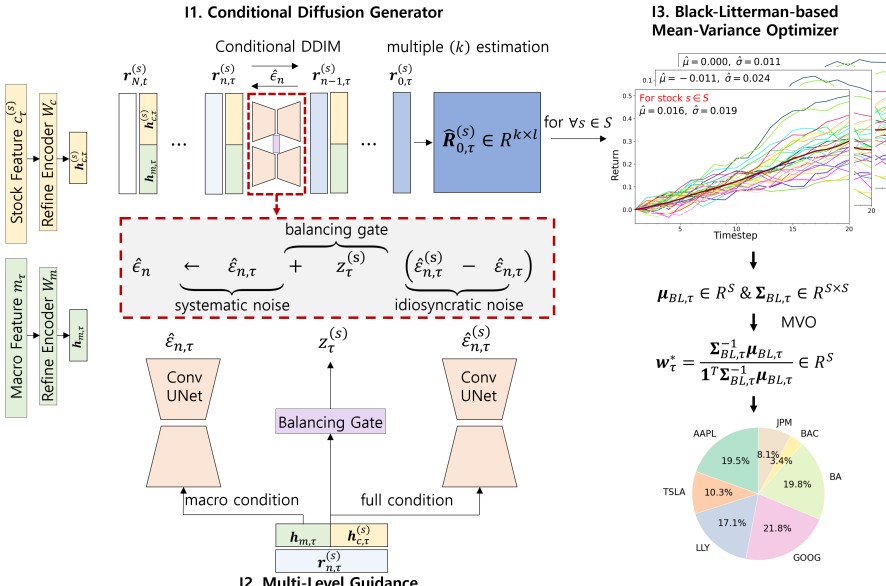

Figure 2: Overview of STABLE. (I1) CDG conditions on macro and firm signals to generate per-stock return segments. (I2) MLG decomposes noise into shared (systematic) and idiosyncratic parts via a learnable gate. (I3) BL–MVO fuses diffusion views with a rolling prior to produce regime-aware portfolio weights.

C1 **Regime shift.** How can we accurately estimate future time series in markets whose regimes keep changing?

C2 **Multi-level factors.** How can we separate, at each time step for each stock, the influence of macro-level factors from stock-level factors?

C3 **Uncertainty shifts.** How can we maintain robust portfolio performance when the certainty of per-step estimates differs over time?

We address the above challenges with the following ideas.

I1 **Conditional diffusion generator (CDG, Sec. 3.3).** We synthesize regime-aware return paths by conditioning diffusion on macro-regime features and stock-specific features.

I2 **Multi-Level Guidance (MLG, Sec. 3.4).** We adapt a guidance modeling framework to financial domain to estimate a shared systematic noise and a stock-specific idiosyncratic noise with a learnable gate that adjusts their relative importance over time.

I3 **Black–Litterman–based mean–variance optimizer (BL–MVO, Sec. 3.5).** We combine diffusion-based views with a certainty weighted Black–Litterman update, which yields per-time, per-stock posterior estimates and enables rational, robust allocation.

## 3.3 CONDITIONAL DIFFUSION GENERATOR (CDG)

**Regime-aware sampling.** STABLE estimates return segments with a Denoising Diffusion Implicit Model (DDIM)-based conditional diffusion sampler (see A.2 in Appendix for details of DDIM). Under the standard random-walk view of log returns, price noise is modeled as Gaussian (Fama, 1995). Diffusion models likewise inject Gaussian perturbations in the forward process and learn to remove them in reverse, so the noise assumptions are aligned with our conditional setting. This alignment makes conditional DDIM a natural mechanism to generate stock-return paths conditioned on market and firm states.

Table 1: Macro indices and description. After $\nu$-day rolling normalization and log differencing, raw and processed values compose $m_\tau \in \mathbb{R}^{d_m}$.

| Index | Description |
|---|---|
| Market index | Region-level equity baseline (overall market condition). |
| Dollar index | Currency strength affecting equities via FX channels. |
| U.S. term spread (10y–3m) | Bond-market condition and liquidity measure. |
| VIX index | Forward-looking equity volatility (risk expectation). |
| Gold index | Risk aversion / inflation measure. |

**Inputs and refinement.** At rebalancing time $\tau$, CDG conditions on two inputs that summarize market context and firm identity. The *macro feature* $m_\tau \in \mathbb{R}^{d_m}$ encodes the global regime at time $\tau$ and is refined by a linear layer $W_m \in \mathbb{R}^{d_m \times d}$ into a refined macro condition $h_{m,\tau} = m_\tau^T W_m \in \mathbb{R}^{1 \times d}$. The macro features are described in Table 1. The *corporate-specific feature* $c_\tau^{(s)} \in \mathbb{R}^{d_c}$ for stock $s$ at time $\tau$ concatenates (i) the temporal stock embedding $\beta_\tau^{(s)}$ at $\tau$ (Das & Ghoshal, 2010), and (ii) the last normalized adjusted-close level and daily log returns for $s$. A linear layer $W_c \in \mathbb{R}^{d_c \times d}$ yields a refined stock-level condition $h_{c,\tau}^{(s)} = c_\tau^{(s)T} W_c \in \mathbb{R}^{1 \times d}$, and the refined full condition becomes $h_{f,\tau}^{(s)} = [\, h_{m,\tau} \,\|\, h_{c,\tau}^{(s)} \,] \in \mathbb{R}^{1 \times 2d}$. Given $h_{f,\tau}^{(s)}$, CDG produces a denoised length-$\ell$ return segment $\widehat{r}_{0,\tau}^{(s)} \in \mathbb{R}^\ell$ for stock $s$ at time $\tau$.

**Temporal stock embedding.** We construct part of the stock-specific feature $c_\tau^{(s)}$ using a temporal stock embedding vector $\beta_\tau^{(s)} \in \mathbb{R}^{d_m}$ at time $\tau$. Prior approaches often use static metadata such as sector labels or neural embeddings from price series (Dolphin et al., 2022), but these representations are either fixed over time or fail to reflect macroeconomic regimes.

To address this, we apply Kalman filtering to estimate a time-varying coefficient $\beta_\tau^{(s)}$ with respect to the macro input $m_\tau$. Given the log return $y_\tau^{(s)}$ of stock $s$ as the dependent variable and the macro vector $m_\tau$ as the independent variable, we estimate a contemporaneous macro sensitivity vector $\beta_\tau^{(s)}$ via recursive filtering. This posterior estimate $\beta_\tau^{(s)}$ reflects the stock's embedding at time $\tau$, incorporating all observations up to that point. It serves as a temporal and robust representation that captures regime-aware macro sensitivity of each stock.

**Conditioned DDIM synthesis.** We first define the base conditioned denoiser. The multi-level conditioning and its decomposition are detailed in Section 3.4, and the DDIM notation is defined in Section A.1. Let $r_{n,\tau}^{(s)} \in \mathbb{R}^\ell$ be the reverse–chain state for stock $s$ at time $\tau$ and DDIM step $n$ ($n = N, \ldots, 1, 0$). The denoiser predicts noise $\widehat{\epsilon}$ with $r_{n,\tau}^{(s)}$ and $h_{f,\tau}^{(s)}$, and DDIM updates

$$\widehat{r}_{0,\tau}^{(s)} = \frac{r_{n,\tau}^{(s)} - \sqrt{1 - \bar{\alpha}_n}\, \widehat{\epsilon}}{\sqrt{\bar{\alpha}_n}}, \qquad r_{n-1,\tau}^{(s)} = \sqrt{\bar{\alpha}_{n-1}}\, \widehat{r}_{0,\tau}^{(s)} + \sqrt{1 - \bar{\alpha}_{n-1}}\, \widehat{\epsilon}, \qquad (\eta = 0)$$

**Training objective.** STABLE minimizes the diffusion mean-squared error (MSE) across all stocks $s \in \mathcal{S}$, rebalancing times $\tau \in \mathcal{T}$, and DDIM steps $n \in \mathcal{N}$. To mitigate overfitting, we add an $\ell_2$ penalty on the parameters with hyper parameter $\beta$.

$$\min_\theta \mathcal{L}(\theta) = \mathbb{E}_{s,\tau,n,\epsilon} \| \epsilon - \widehat{\epsilon} \|_2^2 + \beta \| \theta \|_2^2. \tag{1}$$

Since $\epsilon \sim \mathcal{N}(0, I)$ and Eq. (1) minimizes $\mathbb{E} \| \epsilon - \widehat{\epsilon} \|_2^2$, asymptotically we have $\widehat{\epsilon} \sim \mathcal{N}(0, I)$.

## 3.4 MULTI-LEVEL GUIDANCE (MLG)

**Noise decomposition and gate.** We use multi-level guidance that decomposes, for each rebalancing time $\tau$ and stock $s$, the guided noise into a shared (macroeconomic) impact and an unshared (firm-specific) impact with a stock-specific balancing gate. This is motivated by two empirical properties of the financial market. First, macro impact varies over time (Mezei & Sarlin, 2014): during

crises macro variables dominate and cross-stock co-movements are pronounced, whereas in calm periods firm-specific signals carry more weight. Second, sensitivities to macro variables differ across stocks (Shiller, 1995). By explicitly encoding this multi-level context in the denoising step, the noise estimation becomes more accurate for both regime-driven and idiosyncratic dynamics.

For stock $s$ at time $\tau$ at DDIM step $n$, define

$$\widehat{\epsilon} := \epsilon_\theta\big(r_{n,\tau}^{(s)}, n, h_{\mathrm{f},\tau}^{(s)}\big) = \underbrace{\widehat{\varepsilon}_{n,\tau}}_{\text{shared (systematic)}} + \overbrace{z_\tau^{(s)}}^{\text{balancing gate}} \underbrace{\Big(\widehat{\varepsilon}_{n,\tau}^{(s)} - \widehat{\varepsilon}_{n,\tau}\Big)}_{\text{firm-specific (unsystematic)}} .$$

$$\widehat{\varepsilon}_{n,\tau} = u_\phi\big(r_{n,\tau}^{(s)}, n, h_{m,\tau}\big), \qquad \widehat{\varepsilon}_{n,\tau}^{(s)} = u_\phi\big(r_{n,\tau}^{(s)}, n, h_{\mathrm{f},\tau}^{(s)}\big), \qquad z_\tau^{(s)} = g_\pi\big(h_{\mathrm{f},\tau}^{(s)}\big) \in [0, z_{\max}].$$

All labeled terms in the equation refer to three well-defined elements: (i) the shared noise term $\widehat{\varepsilon}_{n,\tau}$, (ii) the firm-specific residual $\widehat{\varepsilon}_{n,\tau}^{(s)} - \widehat{\varepsilon}_{n,\tau}$, and (iii) the scalar gate $z_\tau^{(s)}$. The two noise terms are evaluations of the convolutional UNet denoiser $u_\phi$ at DDIM step $n$ with inputs $(r_{n,\tau}^{(s)}, n, h)$. We use $h_{m,\tau}$ to obtain the shared term $\widehat{\varepsilon}_{n,\tau}$ and $h_{\mathrm{f},\tau}^{(s)}$ to obtain the full-condition term $\widehat{\varepsilon}_{n,\tau}^{(s)}$. The gate is produced by a linear map $g_\phi$, which adjusts, at the stock level, the balance between macro impact and micro dynamics. Here $u_\phi$ takes three inputs: the recovered state $r_{n,\tau}^{(s)}$, the step index $n$, and the condition vector $h \in \{h_{m,\tau}, h_{\mathrm{f},\tau}^{(s)}\}$.

**Training objective and optimization.** We train all parameters jointly to minimize the diffusion MSE. The optimization drives the gate $z_\tau^{(s)}$ downward when $h_{\mathrm{f},\tau}^{(s)}$ indicates macro-level synchronization, and upward in decoupling regimes to allocate more weight to the firm-specific residual. We replace $\widehat{\epsilon}$ with $\epsilon_\theta\big(r_{n,\tau}^{(s)}, n, h_{\mathrm{f},\tau}^{(s)}\big)$, and rewrite the diffusion objective in Eq. (1) as

$$\mathcal{L}(\theta) = \mathbb{E}\left\| \epsilon - \epsilon_\theta\big(r_{n,\tau}^{(s)}, n, h_{\mathrm{f},\tau}^{(s)}\big) \right\|_2^2 + \beta \|\theta\|_2^2, \qquad \theta = \{\phi, \pi, W_m, W_c\}. \tag{2}$$

## 3.5 BLACK–LITTERMAN–BASED MEAN–VARIANCE OPTIMIZER (BL–MVO)

**Black–Litterman using generative sampler.** STABLE performs portfolio allocation by feeding its stock-wise time-series estimates into the Black–Litterman (BL) algorithm to obtain an updated posterior and then solving mean–variance optimization (MVO), following the standard approach for incorporating views in the Black–Litterman model (Idzorek, 2007).

**Views from CDG+MLG.** For each stock $s$ at time $\tau$, we generate $k$ guided paths and stack them as $\widehat{R}_{0,\tau}^{(s)} \in \mathbb{R}^{k \times \ell}$, whose $i$-th row constitutes the $i$-th estimation of the denoised sequence $\widehat{r}_{0,\tau}^{(s)}$, denoted as $\widehat{r}_{0,\tau}^{(s,i)}$. From these samples, we first calculate the view mean vector $\mu_{\mathrm{view},\tau} \in \mathbb{R}^S$, where the $s$-th entry is computed as the sample mean of the generated paths:

$$\mu_{\mathrm{view},\tau}^{(s)} = \frac{1}{k} \sum_{i=1}^{k} \bar{r}_{0,\tau}^{(s,i)}, \tag{3}$$

where $\bar{r}_{0,\tau}^{(s,i)}$ represents the cumulative return derived from the sequence $\widehat{r}_{0,\tau}^{(s,i)}$. Using this mean, we construct the view covariance matrix $\Sigma_{\mathrm{view},\tau} \in \mathbb{R}^{S \times S}$ as the unbiased sample covariance of the cumulative-return vectors across the $k$ paths:

$$\Sigma_{\mathrm{view},\tau} = \frac{1}{k-1} \sum_{i=1}^{k} \left( \mathbf{r}_{0,\tau}^{(i)} - \mu_{\mathrm{view},\tau} \right) \left( \mathbf{r}_{0,\tau}^{(i)} - \mu_{\mathrm{view},\tau} \right)^\top. \tag{4}$$

Here, we define the return vector for the $i$-th path as $\mathbf{r}_{0,\tau}^{(i)} = [\bar{r}_{0,\tau}^{(1,i)}, \ldots, \bar{r}_{0,\tau}^{(S,i)}]^\top \in \mathbb{R}^S$ to capture the joint estimation errors across all assets. Finally, we fuse these views with the rolling prior $(\mu_{\mathrm{prior},\tau}, \Sigma_{\mathrm{prior},\tau})$ using prior certainty $\Phi_\tau = \Sigma_{\mathrm{prior},\tau}^{-1}$ and view certainty $\Omega_\tau = \Sigma_{\mathrm{view},\tau}^{-1}$.

**Estimated posterior and allocation.** The BL posterior mean is obtained by minimizing the negative log-posterior in Eq. (5) (first-order condition), and the posterior covariance follows from the

Table 2: Summary of datasets.

| Dataset | Region | #Stocks | Train from | Train cut | Test to |
|---------|--------|---------|-----------|-----------|---------|
| S&P500 | United States | 55 | 2013-01 | 2024-09 | 2025-03 |
| CSI300 | China | 55 | 2013-01 | 2024-09 | 2025-03 |
| EUROSTOXX | Europe | 37 | 2013-01 | 2024-09 | 2025-03 |
| KOSPI200 | South Korea | 44 | 2013-01 | 2024-09 | 2025-03 |

canonical Gaussian form in Eq. (7):

$$\mu_{\mathrm{BL},\tau} = (\Phi_\tau + \Omega_\tau)^{-1}(\Phi_\tau \mu_{\mathrm{prior},\tau} + \Omega_\tau \mu_{\mathrm{view},\tau}), \qquad \Sigma_{\mathrm{BL},\tau} = (\Phi_\tau + \Omega_\tau)^{-1}.$$

Given $(\mu_{\mathrm{BL},\tau}, \Sigma_{\mathrm{BL},\tau})$, the Sharpe-maximizing weight solves Eq. (8) and has the closed form

$$w_\tau^\star = \frac{\Sigma_{\mathrm{BL},\tau}^{-1} \mu_{\mathrm{BL},\tau}}{\mathbf{1}^\top \Sigma_{\mathrm{BL},\tau}^{-1} \mu_{\mathrm{BL},\tau}},$$

which enforces the budget constraint $\mathbf{1}^\top w_\tau = 1$ by normalization.

# 4 EXPERIMENT RESULTS

We perform experiments to answer the following research questions.

Q1 **Investment performance and robustness.** How does STABLE compare with state-of-the-art portfolio-allocation baselines in investment performance and robustness?

Q2 **Estimation accuracy.** Is STABLE's DDIM-conditioned future time-series estimation superior to competing generative models?

Q3 **Stock embedding quality.** How effectively does the Kalman-$\beta$–based stock embedding model each stock's latent state?

## 4.1 EXPERIMENTAL SETTINGS

All experiments run on a workstation with 4 RTX 3080 GPUs.

**Datasets.** To evaluate STABLE fairly across both market-wide and sector-level universes, we build four regional datasets with identical splits. Following the Global Industry Classification Standard (GICS) as of 2025-03-31, we select the top five largest stocks in each of the eleven sectors for S&P500 and CSI300, and the top four per sector for EUROSTOXX and KOSPI200. We exclude stocks whose available histories do not span the entire sample window. Table 2 summarizes the universe and the split period.

**Competitors.** **Q1** aims to demonstrate that STABLE achieves both profitability and robustness in investment performance. We compare against a range of classical and learning-based portfolio allocation methods. CRP (Kelly, 1956) allocates equal weights and rebalances to equality at every decision time. MVO (Markowitz, 1952a) solves the mean–variance program that maximizes expected return for a target variance using recent mean and covariance estimates, producing stable allocations under approximately stationary regimes. MOM (Jegadeesh & Titman, 1993) assigns larger weights to recent winners based on a lookback momentum score and rebalances on a schedule, and is suitable when returns tend to continue for the next few months. DeepTrader (Wang et al., 2021) learns to determine portfolio-level long–short balances from macro features using reinforcement learning. MetaTrader (Niu et al., 2022) classifies the market regime based on macro inputs and switches among CRP, MVO, and MOM accordingly. AlphaMix (Sun et al., 2023) routes market state representations to multiple neural allocators and selects the best action based on a soft gating policy.

**Q2** aims to demonstrate that STABLE 's conditional generation of future time series is accurate. Accordingly, for portfolio allocation (Section 4.2), we evaluate classical allocators and regime-aware RL allocators. For time–series prediction (Section 4.3), we benchmark representative generative forecasters. Diffusion-TS (Yuan & Qiao, 2024) is a diffusion-based time-series model that reconstructs past data and forecasts future trajectories using a spectral loss. AEC-GAN (Wang et al., 2023) augments a GAN with adversarial error correction over an autoregressive backbone to im-

Table 3: Portfolio allocation results on sector-diversified multi-region datasets. Best performance per column in **bold**. RMDD and AVol are expressed in percentage units (%).

| Method | S&P500 (US) | | | CSI300 (China) | | |
|---|---|---|---|---|---|---|
| | ASR ($\uparrow$) | RMDD ($\downarrow$) | AVol ($\downarrow$) | ASR ($\uparrow$) | RMDD ($\downarrow$) | AVol ($\downarrow$) |
| CRP | 0.82 | 8.89 | 14.44 | -0.70 | 10.96 | 18.21 |
| MVO | 1.18 | 9.00 | 21.18 | -0.66 | 13.18 | 25.93 |
| MOM | 0.03 | 11.87 | 17.00 | -0.47 | 15.57 | 28.77 |
| DeepTrader | -0.71 | 13.40 | 15.76 | -1.18 | 13.40 | 18.76 |
| MetaTrader | 1.00 | 10.88 | 16.82 | -1.09 | 19.40 | 24.15 |
| AlphaMix | 0.35 | 9.59 | 13.92 | -0.80 | 9.59 | 19.11 |
| STABLE (proposed) | **1.85** | **7.82** | **13.43** | **-0.41** | **8.85** | **17.17** |

| Method | EUROSTOXX (Europe) | | | KOSPI200 (South Korea) | | |
|---|---|---|---|---|---|---|
| | ASR ($\uparrow$) | RMDD ($\downarrow$) | AVol ($\downarrow$) | ASR ($\uparrow$) | RMDD ($\downarrow$) | AVol ($\downarrow$) |
| CRP | 1.31 | 5.40 | 12.96 | 0.76 | 8.72 | 26.55 |
| MVO | 0.48 | 8.35 | 16.75 | 0.45 | 11.84 | 29.49 |
| MOM | 1.42 | 5.41 | 12.71 | 0.33 | 13.49 | 23.34 |
| DeepTrader | -2.44 | 15.99 | 12.59 | 0.77 | 9.62 | 23.76 |
| MetaTrader | 0.50 | 9.86 | 14.63 | 0.57 | 10.88 | 22.28 |
| AlphaMix | 1.31 | 5.75 | 11.77 | 1.47 | 9.96 | 18.76 |
| STABLE (proposed) | **2.92** | **3.84** | **10.88** | **1.61** | **8.34** | **17.82** |

prove long-horizon accuracy. KoVAE (Naiman et al., 2023) combines a variational autoencoder with Koopman-operator latent dynamics to model linear evolution in the latent space.

**Evaluation metrics.** We report metrics for the two experiments. For portfolio allocation experiment, we use ASR, RMDD, and AVol. Annualized Sharpe Ratio (ASR) uses daily test returns with a zero risk-free rate and is annualized. Relative Maximum Drawdown (RMDD) is the maximum peak-to-trough loss divided by the peak over the test horizon. Annualized Volatility (AVol) is the standard deviation of daily returns and is annualized. For the time-series prediction task, we report mean squared error (MSE) and dynamic time warping (DTW). MSE is computed between the predicted and true return sequences. DTW measures the average temporal alignment cost between the predicted path and the ground-truth trajectory. Across both experiments, we use a fixed global seed to ensure reproducibility.

**Hyperparameters.** We grid-search the original segment length $\ell \in \{5, 10, 20\}$, macro and stock encoder width $\{8, 16, 32\}$, number of reverse DDIM steps $\{20, 30, 50, 80, 100\}$, and the number $k$ of generated paths per stock for BL views $\{20, 30, 50\}$. We also investigate the forward noising steps used during training over $\{100, 200, 400\}$ and the DDIM noise scale $\eta \in \{0, 0.01, 0.1, 0.2\}$. The gate cap $z_{max} \in \{2, 3, 4\}$, the BL prior window length $\nu \in \{60, 120, 250\}$, and the $\ell_2$ regularization weight $\beta \in \{0.001, 0.01, 0.1\}$ are tuned per dataset. Baselines follow official implementations or paper-reported settings. We report the selected hyperparameters for each dataset in Appendix A.8, and provide details on computational transparency and reproducibility in Appendix A.9.

## 4.2 PORTFOLIO MANAGEMENT PERFORMANCE (Q1)

Table 3 reports ASR, RMDD, and AVol across sector-diversified regional universes. STABLE is ranked first on all three metrics in every region, delivering higher risk-adjusted returns together with lower drawdowns and lower volatility. We analyze the results for different regimes in Appendix A.4.

Classical strategies (CRP, MVO, MOM) are sensitive to regime shifts. In our test window with frequent drawdowns and rebounds, they exhibit larger RMDD and higher AVol, which lowers ASR relative to a regime-aware approach. MetaTrader, which switches among the classical strategies using macro cues, inherits the same limitations and shows similar patterns.

Among RL baselines, DeepTrader performs the worst. Its macro-only controller struggles during the "new normal" period with a persistent yield-curve inversion, where macro signals alone are not sufficient to determine long–short balance. AlphaMix is the strongest competitor because it routes

Table 4: Time–series prediction on sector-diversified markets. Lower is better. We report MSE and DTW aggregated over all stocks and rebalancing times. MSE is shown in $\times 10^{-4}$. Normalized DTW is shown in $\times 10^{-3}$. Best per column in **bold**.

| | S&P500 | | CSI300 | | EUROSTOXX | | KOSPI200 | |
|---|---|---|---|---|---|---|---|---|
| Method | MSE ($\downarrow$) | DTW ($\downarrow$) | MSE ($\downarrow$) | DTW ($\downarrow$) | MSE ($\downarrow$) | DTW ($\downarrow$) | MSE ($\downarrow$) | DTW ($\downarrow$) |
| Diffusion-TS | 3.90 | 5.73 | 5.71 | 6.78 | 3.05 | 5.80 | 9.41 | 8.70 |
| AEC-GAN | 4.27 | 6.58 | 4.57 | 6.13 | 3.70 | 7.40 | 10.18 | 9.28 |
| KoVAE | 4.58 | 5.93 | 5.46 | 7.28 | 2.61 | 5.43 | 9.83 | 8.73 |
| STABLE (proposed) | **3.51** | **5.62** | **3.89** | **6.09** | **2.49** | **4.78** | **8.15** | **8.67** |

Table 5: Top-5 most similar stocks (by Euclidean distance) returned by our dynamic embeddings for four query snapshots. For TSLA on 2024-12-31, the global context most strongly reflects the prices of NVDA, AVGO, AAPL, MSFT, and GOOGL.

| Query | Top 1 | Top 2 | Top 3 | Top 4 | Top 5 |
|---|---|---|---|---|---|
| TSLA @ 2021-06-28 | AAPL | AVGO | MA | META | ECL |
| TSLA @ 2024-12-31 | NVDA | AVGO | AAPL | MSFT | GOOGL |
| BAC @ 2021-06-28 | JPM | WELL | WFC | DUK | MCD |
| BAC @ 2024-12-31 | WFC | JPM | ECL | LIN | APD |

among policies by market state, yet it does not model time-varying stock-specific properties. In contrast, STABLE conditions on temporal Kalman $\beta$ embeddings and adapts to stock-level regime changes, which explains the consistent gains in ASR, RMDD, and AVol.

## 4.3 TIME–SERIES PREDICTION ACCURACY (Q2)

We assess the accuracy of conditional future return sequences predicted by STABLE. At each rebalancing time we generate $k$ trajectories per stock. We report two error metrics aggregated over all stocks and times: MSE of the mean-of-$k$ forecaster and the average DTW distance across the $k$ generated paths. Lower is better for both.

Table 4 summarizes mean squared error (MSE) and dynamic time warping (DTW) across all four markets. STABLE achieves the lowest MSE and DTW in every market, indicating the most accurate conditional estimation of future return segments.

Diffusion-TS is the strongest baseline but remains behind STABLE. It is designed for broad generalization and does not modulate the importance of conditions at the individual-stock level. In contrast, STABLE decomposes the noise into *systematic* noise that captures market-wide regularities and *idiosyncratic* noise that captures firm-specific patterns. This dual modeling allows the relative weight of conditions to vary by stock and time, which improves sequence alignment and reduces prediction error. Furthermore, we evaluate the covariance estimation performance derived from these return sequences against non-generative and deep generative models in Appendix A.5. We also present a stylized facts validation and goodness-of-fit tests on the distributions of returns generated by these models in Appendix A.6.

## 4.4 STOCK EMBEDDING QUALITY (Q3)

To assess embedding quality, we perform a nearest-neighbor analysis on representative U.S. stocks in our universe. For each query date, we retrieve the most similar embeddings and verify whether the matches share sector category or display similar price dynamics, as expected when the embedding faithfully encodes stock identity.

Table 5 shows the five closest neighbours from the dynamic embeddings for TSLA (Tesla) and BAC (Bank of America) at two snapshots (2021 and 2024). In 2021 TSLA is nearest to Big-Tech stocks such as AAPL and AVGO. By late 2024 its closest neighbours shift to AI-focused firms like NVDA and MSFT. This shift illustrates how the embeddings track the market's AI boom. BAC stays close to JPM and WFC at both times, confirming a stable financial-sector relationship that our method captures over time.

## 5 CONCLUSION

We propose STABLE, a regime-adaptive portfolio framework that unifies three modules. First, the *Conditional Diffusion Generator (CDG)* uses market regime and stock identity as conditions, and enables accurate per-stock time-series estimation. Second, the *Multi-Level Guidance (MLG)* estimates, for each stock and time, how strongly macro and micro impacts drive the denoising process through a learnable gate. Third, the *Black–Litterman–based Mean–Variance Optimizer (BL–MVO)* incorporates sampling certainty into view formation and produces rational and robust allocations. On real-world sector-diversified, multi-region stock market datasets, STABLE outperforms competitors portfolio allocation and estimation tasks. For portfolio allocation, Annualized Sharpe Ratio improves by up to **122.9%**, relative maximum drawdown decreases by up to **1.56%**p, and annualized volatility decreases by up to **7.56%**. For future time-series estimation, STABLE reduces MSE by up to **15.7%** and DTW by up to **13.8%** against the best competitor. Future works include extending our method to exploit more rich features including texts for macro and stock features.

ACKNOWLEDGMENTS

This work was supported by DeepTrade Technologies Inc. U Kang is the corresponding author.

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

# A  APPENDIX

## A.1  SYMBOLS

The symbols used in this paper is summarized in Table 6

Table 6: **Notation summary of STABLE.**

| Symbol | Description |
| --- | --- |
| $S$ | Number of stocks in the universe. |
| $\ell$ | Length of the per-stock return segment reconstructed by the diffusion model. |
| $r_\tau^{(s)} \in \mathbb{R}^\ell$ | Next-period log-return segment of stock $s$ (length $\ell$). |
| $r_{n,\tau}^{(s)} \in \mathbb{R}^\ell$ | Reverse-chain state at step $n$ ($r_{0,\tau}^{(s)}$ clean; $r_{N,\tau}^{(s)}$ noisy). |
| $h_{m,\tau} \in \mathbb{R}^d$ | Macro representation from $m_\tau$. |
| $h_{c,\tau}^{(s)} \in \mathbb{R}^d$ | Stock-specific representation obtained from $c_\tau^{(s)}$. |
| $h_{f,\tau}^{(s)} \in \mathbb{R}^{2d}$ | Combined representation, $[\, h_{m,\tau} \,\|\, h_{c,\tau}^{(s)} \,]$. |
| $\alpha_n, \beta_n, \bar{\alpha}_n$ | Diffusion schedule ($\alpha_n = 1 - \beta_n$, $\bar{\alpha}_t = \prod_{i=1}^n \alpha_i$). |
| $\mu_{\mathrm{prior},\tau} \in \mathbb{R}^S$ | Prior mean vector at time $\tau$. |
| $\Sigma_{\mathrm{prior},\tau} \in \mathbb{R}^{S \times S}$ | Prior covariance. |
| $\mu_{\mathrm{view},\tau} \in \mathbb{R}^S$ | View mean vector. |
| $\Sigma_{\mathrm{view},\tau} \in \mathbb{R}^{S \times S}$ | View covariance. |
| $\mu_{\mathrm{BL},\tau} \in \mathbb{R}^S, \Sigma_{\mathrm{BL},\tau} \in \mathbb{R}^{S \times S}$ | Black–Litterman posterior mean and covariance. |
| $\pi$ | Parameters of the balancing gate $g_\pi$. |
| $\phi$ | Parameters of the UNet denoiser $u_\phi$. |

## A.2 DENOISING DIFFUSION IMPLICIT MODELS USING GUIDANCE MODELING

**Denoising Diffusion Implicit Models (DDIM).** DDIM provides a non-Markovian implicit reverse sampler that preserves the training-time marginals while allowing a much coarser (fewer-step) discretization of the reverse process than DDPM. This yields faster inference and, under fixed conditions, more consistent denoised sequences. These properties are crucial for regime-aware forecasting at each decision time $\tau$.

**DDIM training objective.** Training follows the standard noise-prediction objective used by DDPM. For stock $s$ and time $\tau$, let $r_{n,\tau}^{(s)} \in \mathbb{R}^l$ be the noisy return sequence at DDIM reverse step $n$, and let the condition be either a macro vector $m_\tau \in \mathbb{R}^{d_m}$ or a per-stock vector $c_\tau^{(s)} \in \mathbb{R}^{d_c}$. A denoiser $\epsilon_\phi$ predicts the injected Gaussian noise:

$$\ell^\epsilon = \mathbb{E}_{s,\tau,n,\epsilon}\Big[\big\|\epsilon - \epsilon_\phi\big(r_{n,\tau}^{(s)},\, n,\, \mathrm{cond}\big)\big\|^2\Big], \quad \mathrm{cond} \in \{m_\tau,\, c_\tau^{(s)}\}.$$

At test period, DDIM replaces the Markovian reverse chain with a non-Markovian update, so that the same trained $\epsilon_\phi$ can be sampled in substantially fewer steps while keeping faithful marginals.

**Guidance modeling.** Guidance modeling aims to strengthen conditional fidelity by interpolating noise predictions from the same network. We use

$$\epsilon_u = \epsilon_\theta(r_n, n, \mathrm{uncond}), \qquad \epsilon_c = \epsilon_\theta(r_n, n, \mathrm{cond}), \qquad \widehat{\epsilon} = \epsilon_u + z\,(\epsilon_c - \epsilon_u),\ z \geq 0.$$

The scalar $z$ controls the trade-off: larger $z$ pushes sampling toward the conditional mode, while the shared-parameter $\epsilon_u$ keeps samples realistic. The term $(\varepsilon_c - \varepsilon_u)$ acts as a conditional residual that steers the denoising direction toward the conditional mode, while $\varepsilon_u$ regularizes the step toward high-likelihood regions of the unconditional data distribution. Despite fewer reverse steps (DDIM), this residual-plus-regularizer view improves conditional alignment without sacrificing realism.

## A.3 BLACK–LITTERMAN–BASED OPTIMIZATION

Black–Litterman (BL) updates a baseline prior for asset returns with investor views to form a posterior. Given this estimated posterior, BL performs mean–variance–optimal (MVO) portfolio allocation. The update balances the prior and the views by their certainties: highly certain views tilt the posterior toward the views, while low-certainty views keep it close to the prior.

At time $\tau$, let $S$ denote the number of investable stocks in the universe. We use two prior quantities: the prior mean $\mu_{\mathrm{prior},\tau} \in \mathbb{R}^S$ and the prior covariance $\Sigma_{\mathrm{prior},\tau} \in \mathbb{R}^{S \times S}$. Under MVO, the market–equilibrium portfolio weight implied by moments $(\mu, \Sigma)$ is proportional to $\Sigma^{-1}\mu$, which maximizes the expected risk-adjusted return. To anchor this reference in a simple way, we fix the

equilibrium benchmark to the equal-weight portfolio $w_{\text{eq}} \in \mathbb{R}^S$ with entries $1/S$ and define

$$\mu_{\text{prior},\tau} = \Sigma_{\text{prior},\tau} w_{\text{eq}}, \qquad \Phi_\tau = \Sigma_{\text{prior},\tau}^{-1}.$$

This construction makes $w_{\text{eq}}$ coincide with the MVO direction under the prior and defines certainty via the precision $\Phi_\tau$.

Let the per-asset view be $(\mu_{\text{view},\tau}, \Sigma_{\text{view},\tau})$ with view certainty $\Omega_\tau = \Sigma_{\text{view},\tau}^{-1}$.

**Posterior mean via MAP.** Our goal is the posterior mean of returns $\mu_{\text{BL},\tau}$ that balances the prior $\mu_{\text{prior},\tau}$ and the view $\mu_{\text{view},\tau}$ according to their certainties. Under Gaussian prior and view, the maximum a posteriori (MAP) estimator is obtained by minimizing the negative log-posterior (constants omitted):

$$J(\mu) = \tfrac{1}{2}(\mu - \mu_{\text{prior},\tau})^\top \Phi_\tau (\mu - \mu_{\text{prior},\tau}) + \tfrac{1}{2}(\mu - \mu_{\text{view},\tau})^\top \Omega_\tau (\mu - \mu_{\text{view},\tau}). \tag{5}$$

Setting the gradient to zero yields the closed-form solution

$$\mu_{\text{BL},\tau} = (\Phi_\tau + \Omega_\tau)^{-1}\big(\Phi_\tau \mu_{\text{prior},\tau} + \Omega_\tau \mu_{\text{view},\tau}\big). \tag{6}$$

**Posterior covariance via canonical form.** Expanding (5) yields

$$J(\mu) = \tfrac{1}{2}\mu^\top H_\tau \mu - \mu^\top \eta_\tau + \text{const}, \quad H_\tau = \Phi_\tau + \Omega_\tau, \quad \eta_\tau = \Phi_\tau \mu_{\text{prior},\tau} + \Omega_\tau \mu_{\text{view},\tau}.$$

Hence

$$p(\mu \mid \text{prior, view}) \propto \exp\big(-J(\mu)\big) = \exp\big(-\tfrac{1}{2}\mu^\top H_\tau \mu + \mu^\top \eta_\tau\big),$$

which is the canonical form of a multivariate Gaussian. Therefore,

$$\Sigma_{\text{BL},\tau} = H_\tau^{-1} = (\Phi_\tau + \Omega_\tau)^{-1}. \tag{7}$$

**MVO with the BL posterior.** With the Black–Litterman posterior moments $(\mu_{\text{BL},\tau}, \Sigma_{\text{BL},\tau})$, the goal is to maximize the Sharpe ratio

$$w_\tau^\star \in \arg \max_{\mathbf{1}^\top w_\tau = 1} \frac{w_\tau^\top \mu_{\text{BL},\tau}}{\sqrt{w_\tau^\top \Sigma_{\text{BL},\tau} w_\tau}}. \tag{8}$$

Let $a = \Sigma_{\text{BL},\tau}^{1/2} w_\tau$ and $b = \Sigma_{\text{BL},\tau}^{-1/2}\mu_{\text{BL},\tau}$. Then

$$\frac{w_\tau^\top \mu_{\text{BL},\tau}}{\sqrt{w_\tau^\top \Sigma_{\text{BL},\tau} w_\tau}} = \frac{a^\top b}{\|a\|}.$$

By the Cauchy–Schwarz inequality, $\frac{a^\top b}{\|a\|} \le \|b\|$, with equality iff $a$ is colinear with $b$. Thus the maximizing direction satisfies $a = \lambda b$ for some $\lambda > 0$, i.e.,

$$w_\tau = \lambda \Sigma_{\text{BL},\tau}^{-1}\mu_{\text{BL},\tau}.$$

Finally, enforce the budget constraint $\mathbf{1}^\top w_\tau = 1$ to fix the scale:

$$\lambda = \frac{1}{\mathbf{1}^\top \Sigma_{\text{BL},\tau}^{-1}\mu_{\text{BL},\tau}}, \qquad w_\tau^\star = \frac{\Sigma_{\text{BL},\tau}^{-1}\mu_{\text{BL},\tau}}{\mathbf{1}^\top \Sigma_{\text{BL},\tau}^{-1}\mu_{\text{BL},\tau}}.$$

## A.4 PORTFOLIO MANAGEMENT PERFORMANCES OVER MULTIPLE MARKET REGIMES

We evaluate whether STABLE sustains superior risk-adjusted portfolio performance across heterogeneous market regimes and regions. Predictive performance for financial time series often varies with the dataset and the sample window. To verify consistency, we measure results across disjoint periods and markets.

Building on the recent-period results in Section 4.2, we present experimental results for two contrasting market regimes, excluding the recent period. The first regime, the **COVID-19 crisis** (2019-09-01 to 2020-03-31), is a global stock market crisis characterized by an enormous crash and extreme volatility. The second regime is the **Zero Interest Rate Policy (ZIRP)** period (2020-04-01 to 2022-03-31). Following the crisis, global quantitative easing policies kept bond yields and credit spreads near zero, fueling a global stock market rally.

**Results.** STABLE attains the best ASR, RMDD, and AVol in almost all markets across the two contrasting regimes (Tables 7 and 8). In the COVID-19 window (Table 7), STABLE shows strong crisis-resilience. For example, it improves ASR to **1.61** in the US (from 0.07 for MOM) and reduces RMDD significantly. An exception is South Korea (KOSPI200), where STABLE's ASR (-1.30) is

Table 7: **Portfolio performance during COVID-19 (2019-09-01 to 2020-03-31).** Best performance per column in **bold**. RMDD and AVol are in percentage units (%).

| | S&P500 (US) | | | CSI300 (China) | | |
|---|---|---|---|---|---|---|
| **Method** | ASR ($\uparrow$) | RMDD ($\downarrow$) | AVol ($\downarrow$) | ASR ($\uparrow$) | RMDD ($\downarrow$) | AVol ($\downarrow$) |
| CRP | -0.69 | 35.63 | 51.59 | -0.17 | 16.28 | 26.09 |
| MVO | -0.62 | 32.34 | 49.47 | -0.05 | 14.91 | 25.36 |
| MOM | 0.07 | 26.64 | 46.58 | -1.00 | 9.64 | 24.72 |
| DeepTrader | -0.32 | 25.53 | 38.09 | 0.41 | 11.71 | 19.15 |
| MetaTrader | 0.05 | 37.05 | 34.65 | 1.09 | 10.70 | 21.66 |
| AlphaMix | -0.65 | 38.69 | 38.79 | -0.35 | 9.70 | 19.40 |
| STABLE (proposed) | **1.61** | **23.77** | **34.10** | **1.18** | **9.20** | **18.50** |

| | EUROSTOXX (Europe) | | | KOSPI200 (South Korea) | | |
|---|---|---|---|---|---|---|
| **Method** | ASR ($\uparrow$) | RMDD ($\downarrow$) | AVol ($\downarrow$) | ASR ($\uparrow$) | RMDD ($\downarrow$) | AVol ($\downarrow$) |
| CRP | -1.74 | 39.97 | 40.92 | **-1.28** | 37.18 | 38.25 |
| MVO | -1.61 | 37.02 | 39.04 | -1.29 | **36.74** | 36.68 |
| MOM | -3.70 | 50.16 | 45.81 | -2.64 | 46.71 | 41.91 |
| DeepTrader | -1.36 | 39.71 | 32.50 | -1.64 | 36.87 | 40.61 |
| MetaTrader | -0.84 | 37.03 | 37.15 | -1.59 | 53.00 | 40.37 |
| AlphaMix | -0.92 | 42.49 | 34.20 | -1.38 | 43.30 | 37.80 |
| STABLE (proposed) | **-0.70** | **28.17** | **30.30** | -1.30 | 37.65 | **31.19** |

similar to CRP (-1.28). This is an expected outcome, as our BL–MVO framework converges to the equal-weight vector $w_{\text{eq}}$ (which CRP represents) when predicted covariances become extremely high, prioritizing stability. Notably, STABLE still achieves the lowest AVol (31.19) in this market, confirming its stable performance.

In the ZIRP window (Table 8), STABLE demonstrates strong performance in the market rally, achieving the highest ASR in all regions. In China (CSI300), while its AVol (18.40) is slightly higher than MVO's (17.63), it secures the best ASR (1.67), indicating superior risk-adjusted returns. These outcomes are consistent with the regime-shift-tolerant design of STABLE. The diffusion sampler generates regime-aware paths and the MLG mechanism separates macro impact from firm-specific effects at the stock and time level. As noted for the KOSPI200 case, within BL–MVO, as the view precision $\Omega_\tau = \Sigma_{\text{view},\tau}^{-1}$ decreases (e.g., in high uncertainty), the posterior places greater weight on the prior and the allocation approaches the equal-weight vector $w_{\text{eq}}$, which explains the observed stability during extreme episodes.

## A.5 INDEPENDENT TEST OF COVARIANCE ESTIMATION PERFORMANCE

We isolate the covariance estimation to assess whether the diffusion-derived covariance of STABLE delivers meaningful gains relative to simple prediction models and deep generative forecasters. The goal is to test the covariance view in isolation because risk-adjusted portfolio allocation is driven by risk control, which depends on accurate covariance estimation. We conduct this experiment over multiple regimes including the periods examined in Section 4.2 and Section A.4.

In this experiment STABLE 's CDG with MLG generates per-stock return paths and forms the return view $\mu_{\text{view},\tau}$. To decouple the contributions of mean and covariance estimation, we fix $\mu_{\text{view},\tau}$ from STABLE for all methods and replace only the covariance view $\Sigma_{\text{view},\tau}$. Specifically, for each alternative generative model we sample $k$ paths per asset and compute $\Sigma_{\text{view},\tau}$ via Eq. (4) using those paths; for the simple prediction models we directly use their covariance forecast. BL–MVO then solves for the portfolio weights given the shared $\mu_{\text{view},\tau}$ and the method-specific $\Sigma_{\text{view},\tau}$. All settings including datasets and data split follow those of Section 4.2 and Section A.4. In these settings STABLE as the covariance estimator shows consistent improvements compared with all alternatives. We report ASR, RMDD, and AVol on the four regional universes.

**Simple generative forecasters.** We compare STABLE against simple prediction models that include a regression model, a deep learning model, and a tree-based model, each producing

Table 8: **Portfolio performance during the ZIRP window (2020-04-01 to 2022-03-31).** Best performance per column in **bold**. RMDD and AVol are in percentage units (%).

| Method | S&P500 (US) | | | CSI300 (China) | | |
|---|---|---|---|---|---|---|
| | ASR ($\uparrow$) | RMDD ($\downarrow$) | AVol ($\downarrow$) | ASR ($\uparrow$) | RMDD ($\downarrow$) | AVol ($\downarrow$) |
| CRP | 2.16 | 9.46 | 14.71 | 1.15 | 14.37 | 18.24 |
| MVO | 2.50 | 7.84 | 13.94 | 1.09 | 15.63 | **17.63** |
| MOM | -0.19 | 30.17 | 23.03 | -0.01 | 18.19 | 23.20 |
| DeepTrader | -1.93 | 58.13 | 19.44 | 0.68 | 16.57 | 18.96 |
| MetaTrader | 1.41 | 14.38 | 21.44 | 1.36 | 19.76 | 21.77 |
| AlphaMix | 1.60 | 12.10 | 16.70 | 1.27 | 16.40 | 19.30 |
| STABLE (proposed) | **2.58** | **7.14** | **13.50** | **1.67** | **14.01** | 18.40 |

| Method | EUROSTOXX (Europe) | | | KOSPI200 (South Korea) | | |
|---|---|---|---|---|---|---|
| | ASR ($\uparrow$) | RMDD ($\downarrow$) | AVol ($\downarrow$) | ASR ($\uparrow$) | RMDD ($\downarrow$) | AVol ($\downarrow$) |
| CRP | 1.09 | 19.93 | 18.97 | 1.71 | 14.90 | 17.11 |
| MVO | 1.18 | 19.64 | 18.59 | 1.75 | 14.40 | 17.94 |
| MOM | 0.46 | 25.10 | 19.20 | 0.25 | 19.40 | 23.00 |
| DeepTrader | -1.05 | 46.23 | 22.03 | -0.94 | 44.71 | 21.50 |
| MetaTrader | 1.04 | 20.03 | 22.76 | 1.36 | 19.76 | 21.77 |
| AlphaMix | 1.04 | 20.67 | 20.66 | 1.50 | 18.90 | 19.50 |
| STABLE (proposed) | **1.83** | **16.17** | **17.12** | **1.87** | **14.03** | **16.14** |

a full covariance forecast over time. For the neural and tree models we follow established volatility-forecasting settings for LSTM (Bucci, 2020) and LightGBM (Zhang, 2022). Both predict the lower-triangular Cholesky factor $L_\tau$ of the covariance, which reduces the prediction target and improves computational efficiency. The simple prediction models are as follows.

1. **DCC–GARCH(1,1)** (Engle, 2002). A multivariate GARCH that captures time-varying covariances through univariate volatility updates coupled with a dynamic correlation recursion. The notation $(1,1)$ denotes one lag of the innovation and one lag of the conditional variance. It is a widely used regression baseline that reflects evolving correlations.

2. **LSTM–Cholesky** (Nelson et al., 2017). A vanilla LSTM uses the macro sequence in Table 1 over the last $\nu$ business days and predicts the entries of $L_\tau$. The diagonal is constrained positive via a softplus map. The loss is the Gaussian negative log-likelihood on $L_\tau$.

3. **LightGBM–Cholesky** (Ke et al., 2017). We train one gradient-boosted tree regressor per entry $L_{ij,\tau}$. Inputs include the most recent values from the previous rebalance $L_{ij,\tau-\ell}, L_{ii,\tau-\ell}, L_{jj,\tau-\ell}$ together with macro features aggregated over the last $\nu$ business days from Table 1.

**Deep generative forecasters.** We also include diffusion- and autoencoding-based deep generative models (**Diffusion-TS** (Yuan & Qiao, 2024), **AEC–GAN** (Wang et al., 2023), and **KoVAE** (Naiman et al., 2023)) as covariance forecasters by sampling $k$ paths per asset and taking the sample covariance at each $\tau$. STABLE forms $\Sigma_{\text{view},\tau}$ directly from its conditional diffusion paths with multi-level guidance.

**Results.** Tables 9, 10, and 11 summarize portfolio outcomes under the fixed-return setting across multiple regimes. In general, using STABLE for the covariance view achieves the best performance across all metrics in almost every regime and region. However, in South Korea during the COVID-19 period, where STABLE previously showed lower performance than CRP, replacing the covariance with a simple regression model DCC–GARCH(1,1) yields the highest performance.

During the COVID-19 crisis (Table 9), replacing STABLE's covariance with other deep generative models leads to severe performance degradation, with the exception of KoVAE in South Korea. As shown in the stylized facts comparison in Section A.6, these models tend to estimate Kurtosis and Skewness close to a normal distribution. Consequently, they fail to account for the extreme tail risks inherent in market crashes, resulting in significantly worsened stability metrics. For instance, in the S&P500 market, STABLE improves ASR by **92.7%** (from 0.84 to 1.61) compared to the best alternative (Diffusion-TS), while reducing annualized volatility by **1.49%p**. Similarly, in the

Table 9: **Portfolio results with a fixed return view from STABLE (CDG+MLG) during the COVID-19 crisis.** We fix the return view $\mu_{\text{view},\tau}$ to the per-stock estimates from STABLE 's CDG+MLG and vary only the covariance view $\Sigma_{\text{view},\tau}$ using each forecaster. BL–MVO then uses $\left(\mu_{\text{view},\tau}, \Sigma_{\text{view},\tau}\right)$ to produce the allocation. Performance of STABLE corresponds to the results in Table 7. Replacing the covariance view of STABLE generally degrades performance, except for South Korea where DCC–GARCH(1,1) outperforms the others. Best per column in **bold**.

| Covariance forecaster | S&P500 (US) | | | CSI300 (China) | | |
|---|---|---|---|---|---|---|
| | ASR (↑) | RMDD (↓) | AVol (↓) | ASR (↑) | RMDD (↓) | AVol (↓) |
| DCC–GARCH(1,1) | -0.48 | 39.28 | 36.62 | 0.33 | 12.48 | 18.39 |
| LSTM–Cholesky | -0.35 | 45.78 | 35.09 | 0.53 | 11.79 | 19.80 |
| LightGBM–Cholesky | -0.15 | 26.35 | 37.66 | 0.46 | 15.60 | 19.11 |
| Diffusion-TS | 0.84 | 26.51 | 35.59 | 0.63 | 12.17 | 21.26 |
| AEC–GAN | -0.73 | 33.86 | 37.78 | 1.02 | 13.29 | 20.58 |
| KoVAE | -1.04 | 41.36 | 39.57 | 0.87 | 14.51 | 21.15 |
| STABLE (proposed) | **1.61** | **23.77** | **34.10** | **1.18** | **9.20** | **18.50** |
| Covariance forecaster | EUROSTOXX (Europe) | | | KOSPI200 (South Korea) | | |
| | ASR (↑) | RMDD (↓) | AVol (↓) | ASR (↑) | RMDD (↓) | AVol (↓) |
| DCC–GARCH(1,1) | -1.38 | 37.18 | 32.86 | **0.01** | **27.60** | **25.54** |
| LSTM–Cholesky | -2.00 | 34.31 | 35.37 | -1.60 | 32.11 | 33.37 |
| LightGBM–Cholesky | -1.89 | 40.69 | 34.41 | -0.75 | 47.56 | 40.22 |
| Diffusion-TS | -0.93 | 36.32 | 35.26 | -1.88 | 42.55 | 34.18 |
| AEC–GAN | -1.00 | 48.82 | 36.75 | -1.49 | 41.02 | 30.93 |
| KoVAE | -2.61 | 46.57 | 39.48 | -1.26 | 39.22 | 41.78 |
| STABLE (proposed) | **-0.70** | **28.17** | **30.30** | -1.30 | 37.65 | 31.19 |

EUROSTOXX market, STABLE reduces RMDD by **6.14%p** (from 34.31% to 28.17%) compared to the strongest baseline (LSTM), demonstrating superior stability during crises.

In the ZIRP period (Table 10), STABLE's covariance estimation yields the superior investment performance across all regions. This dominance aligns with the findings in Section A.6, where the distribution of returns estimated by STABLE exhibits stylized facts highly similar to those of realized market data, enabling effective risk diversification. Notably, in the S&P500 market, STABLE improves ASR by **25.1%** (from 2.06 to 2.58) and reduces RMDD by **2.49%p** compared to the best competitor (AEC–GAN). In the EUROSTOXX market, STABLE achieves **29.5%** higher ASR (1.83 vs 1.41) than the LSTM baseline, confirming that accurate covariance estimation via STABLE translates directly into enhanced risk-adjusted returns.

For the recent period (Table 11), the performance degradation from replacing STABLE's covariance is the lowest in the EUROSTOXX market. This result is consistent with the findings in Table 4, where the deep generative models exhibit their strongest average MSE and DTW scores. Even here, STABLE improves the ASR by **9.8%** (from 2.66 to 2.92) and reduces RMDD by **14.1%** (from 4.47% to 3.84%) compared to the strongest competitor (KoVAE). Conversely, in regions where generative models have higher MSE and DTW scores, such as CSI300 and KOSPI200 (Table 4), replacing STABLE's covariance leads to severe performance degradation. This result aligns with the time-series estimation performance reported in Table 4 and underscores the effectiveness of STABLE's multi-level guidance. By separating systematic impacts from idiosyncratic characteristics, STABLE maintains robust covariance estimation and risk control even in markets where simpler generative models fail.

## A.6    STYLIZED FACTS ANALYSIS AND GOODNESS-OF-FIT TESTS

We conduct a comprehensive analysis to validate the distributional properties of the return segments generated by STABLE and the deep generative models from Section 4.3. We conduct this experiment over multiple regimes including the periods examined in Section 4.2 and A.4. We perform both stylized facts validation and goodness-of-fit tests on the generated segments against the realized return segments. The objective is to verify that the generated segments, particularly from STABLE, capture not only simple time-series similarity but also the key dynamic properties and underlying stochastic structure of realized returns. For stylized facts validation, we conduct statistical tests to

Table 10: **Portfolio results with a fixed return view from STABLE (CDG+MLG) during the ZIRP period.** Performance of STABLE corresponds to the results in Table 8. STABLE consistently achieves the best performance across all regions. Best per column in **bold**.

| Covariance forecaster | S&P500 (US) | | | CSI300 (China) | | |
|---|---|---|---|---|---|---|
| | ASR (↑) | RMDD (↓) | AVol (↓) | ASR (↑) | RMDD (↓) | AVol (↓) |
| DCC–GARCH(1,1) | 1.39 | 20.81 | 24.27 | 1.47 | 16.96 | 27.15 |
| LSTM–Cholesky | 1.77 | 10.19 | 19.30 | 1.30 | 14.64 | 18.80 |
| LightGBM–Cholesky | 1.36 | 18.81 | 21.44 | 1.11 | 20.70 | 24.92 |
| Diffusion-TS | 0.82 | 12.69 | 22.38 | 1.16 | 21.34 | 24.26 |
| AEC–GAN | 2.06 | 9.63 | 18.94 | 1.38 | 15.68 | 21.53 |
| KoVAE | 1.42 | 11.47 | 20.43 | 1.35 | 14.26 | 21.81 |
| STABLE (proposed) | **2.58** | **7.14** | **13.50** | **1.67** | **14.01** | **18.40** |
| | **EUROSTOXX (Europe)** | | | **KOSPI200 (South Korea)** | | |
| Covariance forecaster | ASR (↑) | RMDD (↓) | AVol (↓) | ASR (↑) | RMDD (↓) | AVol (↓) |
| DCC–GARCH(1,1) | 1.11 | 19.17 | 21.71 | -0.62 | 31.78 | 19.14 |
| LSTM–Cholesky | 1.41 | 18.02 | 20.65 | 1.57 | 21.31 | 20.21 |
| LightGBM–Cholesky | 0.95 | 25.92 | 24.48 | 0.50 | 29.03 | 27.70 |
| Diffusion-TS | 0.87 | 22.61 | 27.94 | 1.27 | 16.94 | 23.74 |
| AEC–GAN | 0.78 | 21.48 | 25.56 | 1.78 | 14.25 | 18.90 |
| KoVAE | 0.66 | 23.54 | 26.69 | 1.66 | 15.64 | 21.98 |
| STABLE (proposed) | **1.83** | **16.17** | **17.12** | **1.87** | **14.03** | **16.14** |

Table 11: **Portfolio results with a fixed return view from STABLE (CDG+MLG) during the recent period.** Performance of STABLE corresponds to the results in Table 3. STABLE consistently achieves the best performance across all regions. Best per column in **bold**. RMDD and AVol are in %.

| Covariance forecaster | S&P500 (US) | | | CSI300 (China) | | |
|---|---|---|---|---|---|---|
| | ASR (↑) | RMDD (↓) | AVol (↓) | ASR (↑) | RMDD (↓) | AVol (↓) |
| DCC–GARCH(1,1) | 0.81 | 10.18 | 16.10 | -0.95 | 14.12 | 19.91 |
| LSTM–Cholesky | 0.82 | 10.22 | 16.94 | -1.23 | 13.94 | 20.12 |
| LightGBM–Cholesky | 0.95 | 13.63 | 20.94 | -1.22 | 14.20 | 21.70 |
| Diffusion-TS | 0.53 | 12.68 | 25.85 | -1.01 | 13.58 | 20.57 |
| AEC–GAN | 0.80 | 12.70 | 19.90 | -0.74 | 16.14 | 21.41 |
| KoVAE | 0.90 | 12.36 | 16.13 | -0.68 | 16.60 | 22.52 |
| STABLE (proposed) | **1.85** | **7.82** | **13.43** | **-0.41** | **8.85** | **17.17** |
| | **EUROSTOXX (Europe)** | | | **KOSPI200 (South Korea)** | | |
| Covariance forecaster | ASR (↑) | RMDD (↓) | AVol (↓) | ASR (↑) | RMDD (↓) | AVol (↓) |
| DCC–GARCH(1,1) | 2.53 | 5.31 | 11.78 | 0.79 | 11.09 | 18.34 |
| LSTM–Cholesky | 2.54 | 5.36 | 14.14 | 1.13 | 12.46 | 19.69 |
| LightGBM–Cholesky | 1.52 | 7.12 | 14.66 | 1.18 | 9.60 | 23.10 |
| Diffusion-TS | 2.65 | 5.21 | 14.55 | -1.88 | 21.93 | 24.98 |
| AEC–GAN | 2.41 | 6.11 | 14.94 | -0.82 | 13.09 | 21.13 |
| KoVAE | 2.66 | 4.47 | 14.59 | 0.66 | 21.70 | 26.29 |
| STABLE (proposed) | **2.92** | **3.84** | **10.88** | **1.61** | **8.34** | **17.82** |

determine if the realized kurtosis, skewness, and autocorrelation of volatility exhibit significant differences from the generated values. For goodness-of-fit tests, we perform the Kolmogorov-Smirnov (KS) (Smirnov, 1948) and Anderson-Darling (AD) (Anderson & Darling, 1952) tests.

**Setup.** We apply a stratified bootstrapping procedure for all tests. Let $S$ be the number of stocks and $T$ be the total number of rebalancing timesteps. Let $N = S \times T$ be the total number of realized segments. Let $D \in \mathbb{R}^{N \times \ell}$ be the set of $N$ realized return segments of length $\ell$, and let $V_D \in \mathbb{R}^{N\ell}$ be its flattened 1D vector. Let $\hat{R}$ represent the set of generated returns, structured as an $(N, k, \ell)$ array, where for each of the $N$ conditions $(s, \tau)$, we have $k$ candidate segments of length $\ell$. We perform $M$ bootstrap iterations. In each iteration $i$, we construct a sample set $D_{\text{sample}}^{(i)} \in \mathbb{R}^{N \times \ell}$ by randomly selecting exactly one segment (out of $k$) for each of the $N$ conditions. This sample is then flattened

into a 1D vector $V^{(i)}_{\text{sample}} \in \mathbb{R}^{N\ell}$. This procedure is applied to STABLE, Diffusion-TS, AEC-GAN, and KoVAE.

**Stylized facts validation.** We analyze three key stylized facts: kurtosis, skewness, and volatility clustering. The results are visualized in Figures 13, 14, and 15, respectively. The validation for each statistic is performed as follows:

- **Kurtosis:** We test whether the single kurtosis value calculated from the flattened realized data $V_D \in \mathbb{R}^{N\ell}$ falls within the 95% confidence interval of the distribution formed by the $M$ kurtosis values, where each kurtosis is calculated from a corresponding flattened bootstrap sample $V^{(i)}_{\text{sample}} \in \mathbb{R}^{N\ell}$.

- **Skewness:** We test whether the single skewness value calculated from the flattened realized data $V_D$ falls within the 95% confidence interval of the distribution formed by the $M$ skewness values calculated from each $V^{(i)}_{\text{sample}}$.

- **Volatility clustering:** We measure the lag-1 autocorrelation of absolute returns (ACF(1)) for each $\ell$-length segment. We test whether the mean ACF(1) calculated across all $N$ segments in the realized data set $D \in \mathbb{R}^{N \times \ell}$ falls within the 95% confidence interval of the distribution formed by the $M$ mean ACF(1) values, where each mean $acf_i$ is calculated from the $N$ segments within a bootstrap sample set $D^{(i)}_{\text{sample}} \in \mathbb{R}^{N \times \ell}$.

**Goodness-of-fit tests.** We use the KS and AD tests to assess the similarity between the generated and realized distributions from two perspectives: overall shape (KS) and tail behavior (AD). The results are visualized in Figures 16 and 17.

- **Kolmogorov-Smirnov (KS) Test:** For each of the $M$ bootstrap samples, we perform a two-sample KS test between the flattened realized data $V_D \in \mathbb{R}^{N\ell}$ and the flattened bootstrap sample $V^{(i)}_{\text{sample}} \in \mathbb{R}^{N\ell}$. We compute the KS statistic based on the maximum discrepancy between the two empirical CDFs (Smirnov, 1948). Instead of relying on asymptotic distributions, we calculate the $p$-value using a permutation test with $B = 200$ iterations to ensure robustness against finite sample sizes and non-normal distributions. We then plot the ECDF of these $M$ $p$-values. If the generated samples are statistically indistinguishable from the realized data, this ECDF plot should be close to the diagonal line (i.e., a uniform distribution).

- **Anderson-Darling (AD) Test:** We conduct a similar procedure using the two-sample AD test. The test statistic is computed as a squared, weighted difference between the ECDFs, giving higher weight to tail deviations (Anderson & Darling, 1952). Consistent with the KS test, the significance levels are determined via a permutation test. We again plot the ECDF of the $M$ resulting $p$-values to check for uniformity.

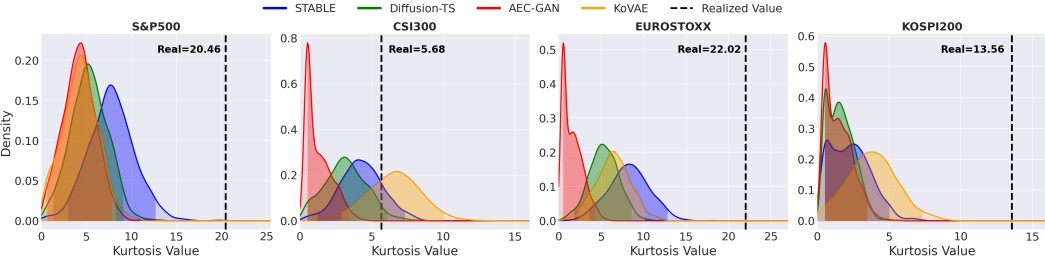

Figure 3: **Stylized Fact Validation (COVID-19): Kurtosis.** Distribution of kurtosis in the COVID-19 regime. Note the extreme realized kurtosis in the S&P500 (20.46). STABLE generally provides robust estimates, though the extreme tails in non-CSI300 markets challenge all models.

**Results.** Overall, STABLE exhibits distribution characteristics the most similar to realized returns across all regimes and stock markets, with the exception of KOSPI200 during the **COVID-19** regime. KoVAE is confirmed to show high similarity for skewness in the **COVID-19** regime. However, KoVAE consistently overestimates the volatility clustering (ACF(1)) of stock returns across all

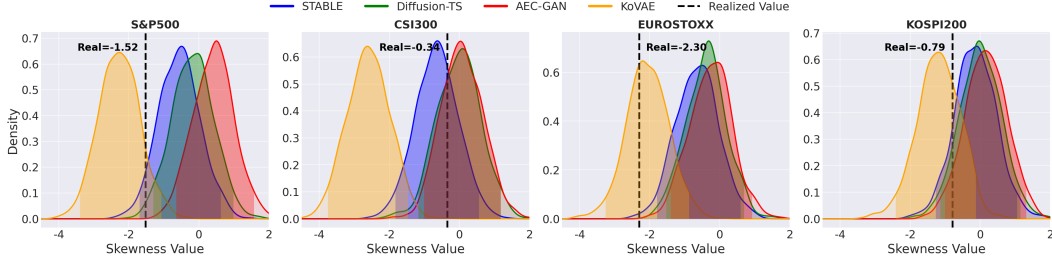

Figure 4: **Stylized Fact Validation (COVID-19): Skewness.** Distribution of skewness in the COVID-19 regime. KoVAE captures the negative skewness in KOSPI200 well, reflecting the market crash dynamics.

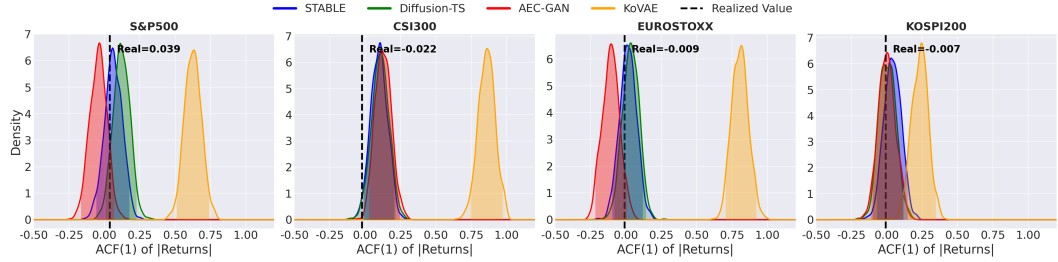

Figure 5: **Stylized Fact Validation (COVID-19): Volatility Clustering (ACF(1)).** Realized values are near zero. STABLE remains consistent, while KoVAE tends to overestimate volatility clustering.

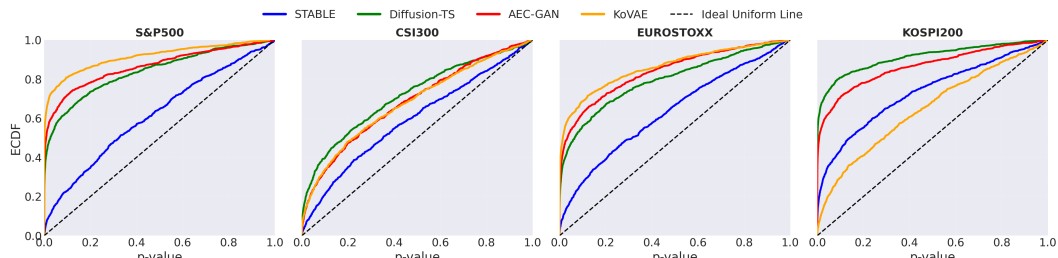

Figure 6: **Goodness-of-Fit (COVID-19): KS Test.** ECDF of $p$-values. STABLE demonstrates superior fit across most markets, with KoVAE showing strength in KOSPI200.

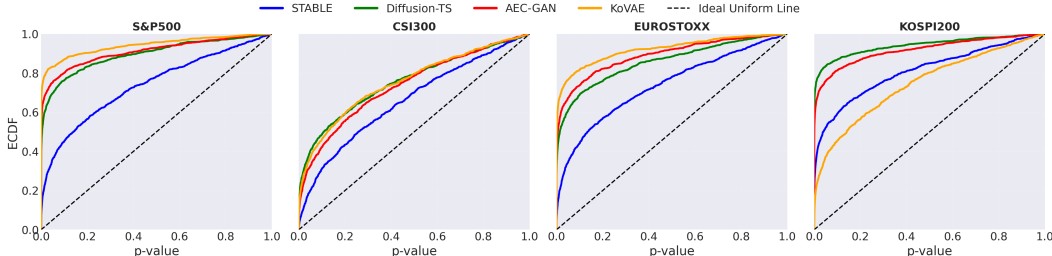

Figure 7: **Goodness-of-Fit (COVID-19): AD Test.** ECDF of $p$-values for the tail-sensitive AD test. The rankings remain consistent with KS tests.

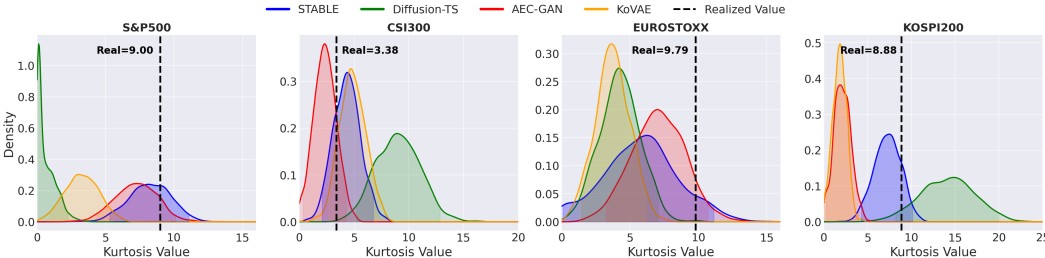

Figure 8: **Stylized Fact Validation (ZIRP): Kurtosis.** Distribution of kurtosis in the ZIRP regime. STABLE successfully captures the kurtosis within confidence intervals in most markets.

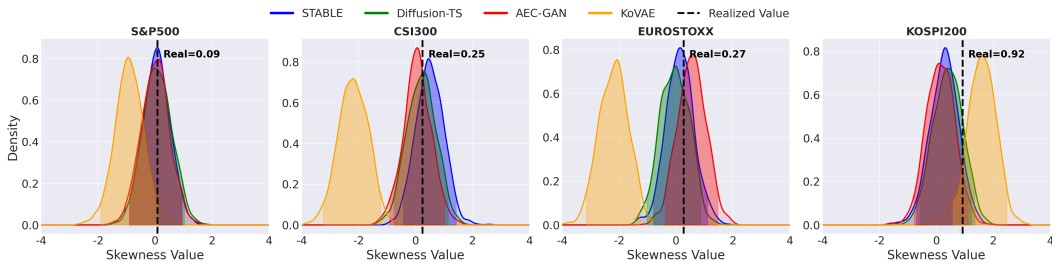

Figure 9: **Stylized Fact Validation (ZIRP): Skewness.** Positive skewness is observed due to the market rally. STABLE and others (except KoVAE) capture this feature well.

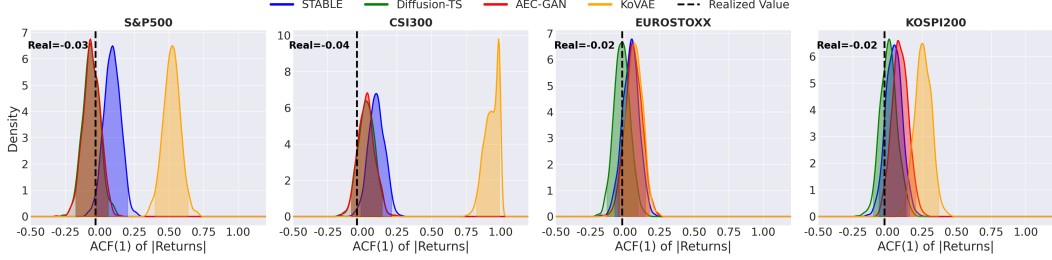

Figure 10: **Stylized Fact Validation (ZIRP): Volatility Clustering (ACF(1)).** STABLE provides accurate estimates near zero, whereas competitors show significant deviations.

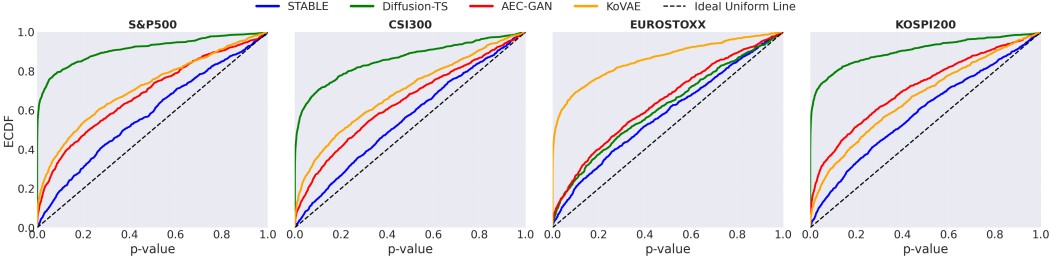

Figure 11: **Goodness-of-Fit (ZIRP): KS Test.** STABLE shows the best fit, aligning closely with the uniform distribution line across all markets.

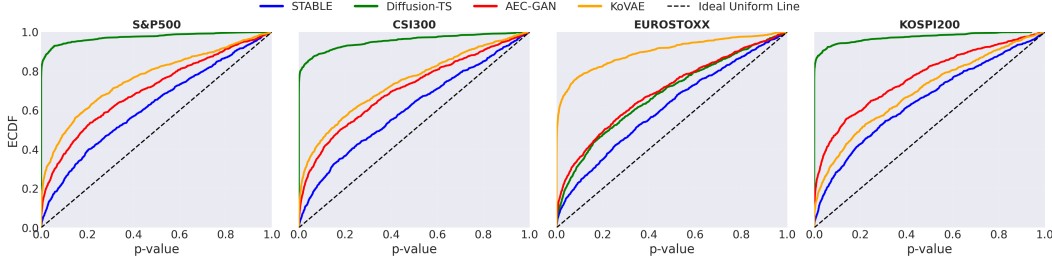

Figure 12: **Goodness-of-Fit (ZIRP): AD Test.** Similar to KS results, STABLE outperforms competitors.

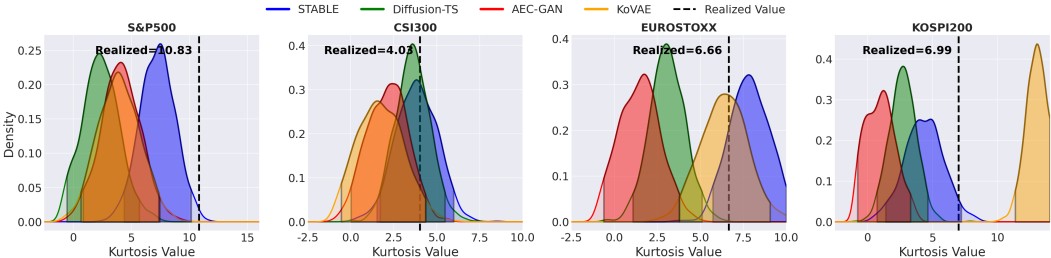

Figure 13: **Stylized Fact Validation (Recent): Kurtosis.** Distribution of kurtosis in the recent regime. The black line is the realized value. All models systematically underestimate the S&P500 kurtosis (10.83). STABLE (blue) provides the closest estimate in CSI300 and KOSPI200, while KoVAE (orange) is closest in EUROSTOXX.

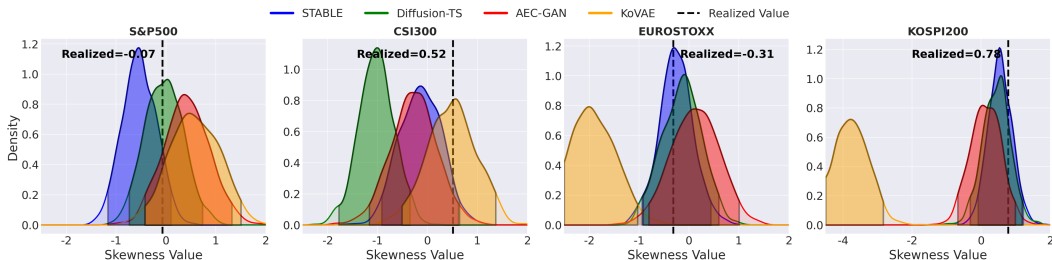

Figure 14: **Stylized Fact Validation (Recent): Skewness.** Distribution of skewness in the recent regime. STABLE (blue) provides the closest estimate in EUROSTOXX and KOSPI200. Diffusion-TS (green) is closest in S&P500, and KoVAE (orange) is closest in CSI300.

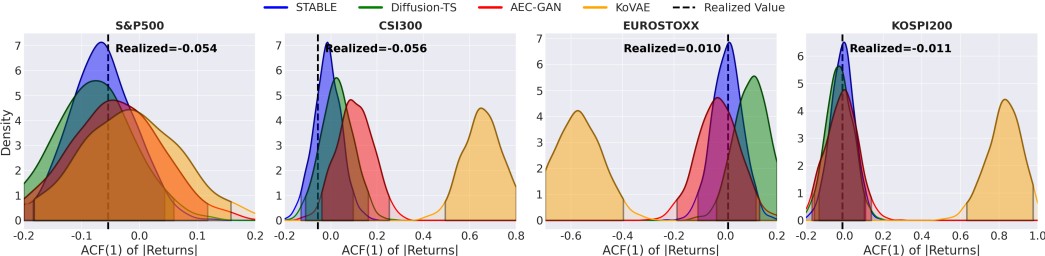

Figure 15: **Stylized Fact Validation (Recent): Volatility Clustering (ACF(1)).** Distribution of the mean ACF(1) in the recent regime. Realized values are near zero. STABLE (blue) provides the closest estimate in all markets. KoVAE (orange) consistently fails, predicting extreme outliers in three markets.

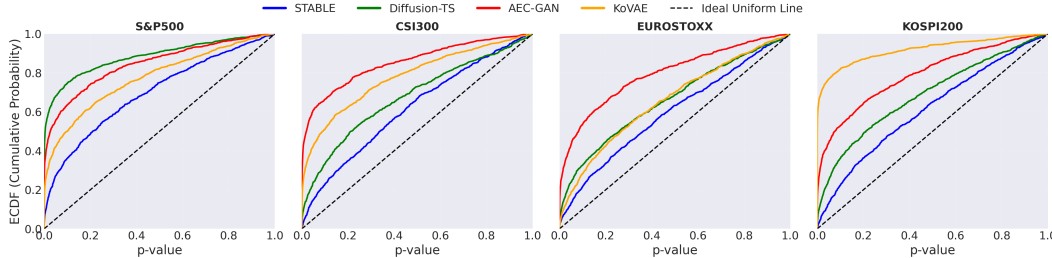

Figure 16: **Goodness-of-Fit (Recent): KS Test.** ECDF of $p$-values. The STABLE curve (blue) is closest to the ideal uniform line (black dash) in all markets, aligning with its top ASR rank in Table 11.

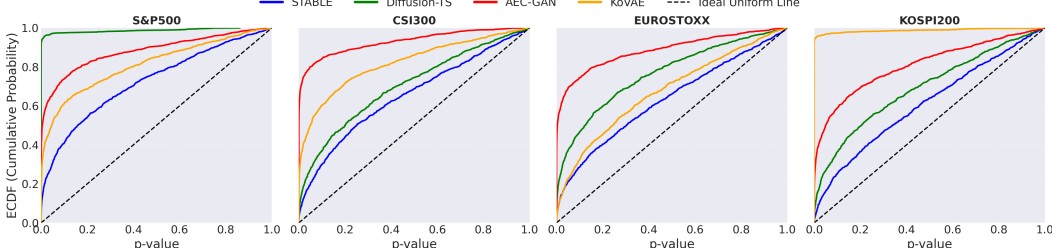

Figure 17: **Goodness-of-Fit (Recent): AD Test.** ECDF of $p$-values for the tail-sensitive AD test. STABLE again provides the best fit in all four markets.

regimes. Consequently, its prediction accuracy for realized returns is not high, resulting in investment performance similar to STABLE in Table 9. Notably, excluding the CSI300 market, the regimes we select exhibit very high kurtosis (specifically, 20.46 for the S&P500 in the **COVID-19** regime). This indicates that the market regimes during our experimental periods contain high volatility due to crashes, surges, and political issues, whereas the CSI300 market has relatively limited tail values likely due to price fluctuation regulations. Additionally, volatility clustering based on ACF(1) is confirmed to be nearly zero across all regimes.

The stylized facts validation results are as follows.

**COVID-19** (Figures 3–5): STABLE demonstrates the most homogeneous stylized facts across most markets. The realized kurtosis in the S&P500 is extremely high (20.46) (Figure 3), which challenges all models. However, STABLE (7.74) provides a distribution closer to the realized value compared to KoVAE (4.30). In KOSPI200, consistent with the results in Table 9, KoVAE is confirmed to generate the return distribution the most similar to reality. In particular, KoVAE's generated skewness (-1.18) (Figure 4) closely matches the realized value (-0.80), accurately capturing the negative skewness during the crash period, whereas STABLE (-0.13) underestimates this asymmetry. Regarding volatility clustering (Figure 5), the realized values are near zero across all markets (e.g., S&P500 0.04), but KoVAE predicts extreme outliers (e.g., S&P500 0.63, CSI300 0.86), showing significant deviation from reality.

**ZIRP** (Figures 8–10): STABLE shows the best stylized facts across all markets. The realized kurtosis in the S&P500 is 9.00 (Figure 8), and STABLE (8.21) successfully captures this within its confidence interval, whereas Diffusion-TS (0.36) significantly underestimates it. A unique characteristic of this regime is the positive skewness observed due to the market rally (e.g., KOSPI200 0.92, CSI300 0.25). Notably, all models except KoVAE include this feature within their confidence intervals. For instance, in KOSPI200 (Figure 9), STABLE (0.27) and Diffusion-TS (0.40) show positive skewness, while KoVAE predicts an excessive value (1.59). In terms of ACF(1) (Figure 10), KoVAE again predicts unrealistic values (e.g., CSI300 0.94), whereas STABLE (0.10) provides a much closer estimate to the realized value (-0.04).

**Recent** (Figures 13–15): The realized kurtosis is high, confirming fat tails. For the S&P500, the realized value (10.83) is an outlier that all models systematically underestimate, although STABLE's distribution is centered the closest (7.27) to this value (Figure 13). The realized skewness is market-dependent. STABLE provides the closest estimate in EUROSTOXX and KOSPI200 (Figure 14). For

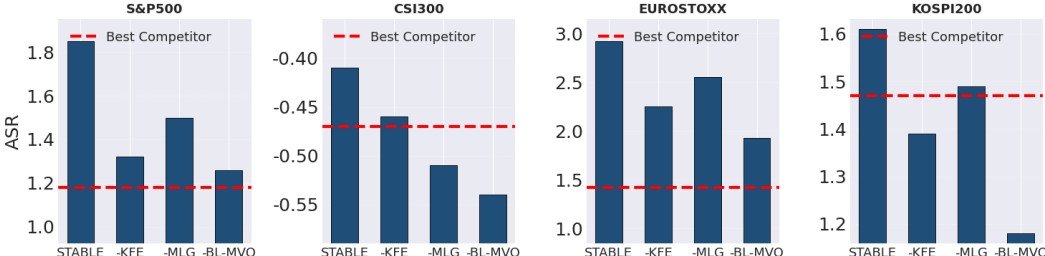

Figure 18: Ablation study of STABLE on ASR. Across all regions the BL–MVO module contributes the most to risk adjusted return.

volatility clustering, STABLE consistently centers its distribution very close to the realized value, whereas KoVAE predicts extreme outliers in three markets (Figure 15).

In terms of the goodness-of-fit, STABLE consistently exhibits the most superior goodness-of-fit in both the KS and AD tests across all evaluated market regimes. In the **COVID-19** regime (Figures 6– 7), STABLE shows the best fit in most markets, with the only exception being the KOSPI200 market where KoVAE demonstrates strength. This robust performance is consistently observed through the **ZIRP** regime (Figures 11–12) and the **Recent** regime (Figures 16–17), where the ECDF curves of the $p$-values for STABLE are unambiguously the closest to the ideal uniform distribution line across all datasets. Overall, the AD test results remain highly consistent with the KS test results, confirming that STABLE outperforms the competitors in accurately modeling the underlying distributions across diverse market conditions.

## A.7 ABLATION STUDY

STABLE is organized into a feature-engineering module and three core modules. The inputs are enriched by the Kalman-filtered stock embedding (KFE), and the core pipeline comprises the Conditional Diffusion Generator (CDG), the Multi-Level Guidance (MLG), and the Black–Litterman–based Mean–Variance Optimizer (BL–MVO). To quantify each component's contribution, we remove or modify one module at a time while keeping all other settings identical to those of Section 4.2. We do not include a variant that removes CDG because allocation requires diffusion-derived views. Accordingly, the ablation variants are as follows.

1. **STABLE without KFE.** The temporal stock embedding $\beta_\tau^{(s)}$ is set to zero, so that the corporate-specific feature $c_\tau^{(s)}$ contains only the last normalized adjusted-close level and daily log returns.

2. **STABLE without MLG.** The noise decomposition and the learnable gate are disabled by fixing the balancing gate to a constant $z_\tau^{(s)} = 0.5$ for all stocks and times. Sampling reduces to conditional DDIM with the full condition $h_{f,\tau}^{(s)}$ without adaptive separation of systematic and idiosyncratic effects.

3. **STABLE without BL–MVO.** The BL update is removed and plug-in MVO uses the diffusion view moments $(\mu_{\text{view},\tau}, \Sigma_{\text{view},\tau})$ directly. This is equivalent to skipping posterior blending.

**Results.** Figure 18 summarizes ASR across the four regional universes. The full STABLE attains the highest ASR in every market. However, removing a module can sometimes results in performance below the best competitor. For instance, in KOSPI200, the variants without KFE or BL–MVO show a lower ASR than AlphaMix (see Table 3). In CSI300, all three ablation variants fall below the ASR of MOM. The degree to which each module contributes varies by dataset. For example, the KFE's influence is larger than MLG's in the S&P500, but their importance is reversed in CSI300. Despite this, BL–MVO consistently plays the most critical role; removing it causes the largest drop in ASR in all markets. This is natural, as the BL–MVO module determines the final portfolio weights by deciding how much to trust the estimates from the other modules.

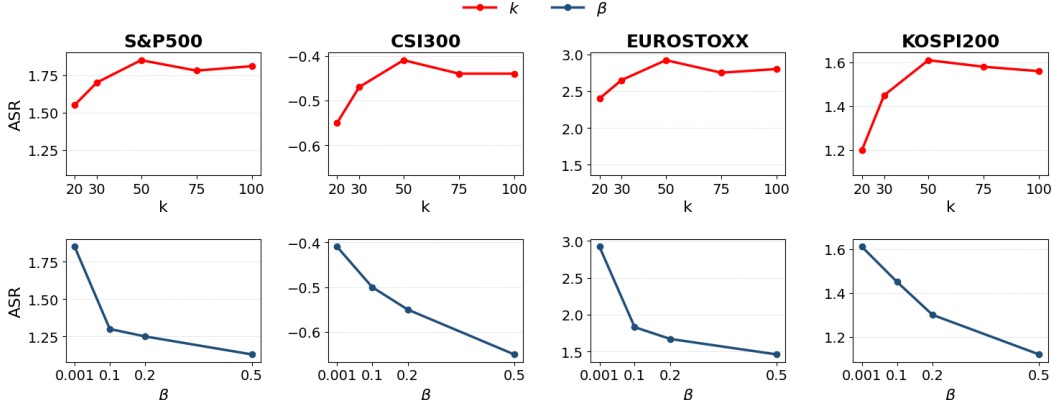

Figure 19: **ASR sensitivity to the most influential hyperparameters.** Top row varies the number $k$ of guided paths. ASR saturates when $k \geq 50$. Bottom row varies the $\ell_2$ weight $\beta$. Values above 0.001 induce excessive regularization and reduce ASR.

Table 12: **Selected hyperparameters by dataset.** Dataset-level grid-search selections for all hyperparameters defined in Section 4.1. Except for the DDIM noise scale $\eta$ which controls sampling stochasticity, all four datasets share the identical optimal values for the remaining hyperparameters.

| Hyperparameter | S&P500 | CSI300 | EUROSTOXX | KOSPI200 |
|---|---|---|---|---|
| Segment length $\ell$ | 20 | 20 | 20 | 20 |
| Macro & stock embedding dim $d$ | 16 | 16 | 16 | 16 |
| # DDIM reverse steps | 30 | 30 | 30 | 30 |
| Number $k$ of paths | 50 | 50 | 50 | 50 |
| # forward diffusion steps | 200 | 200 | 200 | 200 |
| DDIM noise $\eta$ | 0.00 | 0.00 | 0.00 | 0.01 |
| Balancing-gate cap $z_{\max}$ | 2 | 2 | 2 | 2 |
| BL prior window length $\nu$ | 120 | 120 | 120 | 120 |
| $\ell_2$ weight $\beta$ | 0.001 | 0.001 | 0.001 | 0.001 |

## A.8 HYPERPARAMETER ANALYSIS

We measure how STABLE's hyperparameters affect the risk-adjusted return metric ASR. This sensitivity analysis assesses how responsive the method is to hyperparameter choices and informs the generalization of the selected configuration in practice. In particular, we sweep the number $k$ of guided paths per stock and the $\ell_2$ regularization weight $\beta$ because these two hyperparameters the most strongly affect ASR, while fixing all remaining settings to the dataset-wise choices in Table 12.

The ASR sensitivity to $k$ and $\beta$ is visualized in Figure 19. For $k$, $k = 50$ is optimal across regions and the ASR curve saturates beyond this point. For $\beta$, the best value is $\beta = 0.001$ and larger values suppress performance due to over-regularization.

## A.9 COMPUTATIONAL TRANSPARENCY AND REPRODUCIBILITY

For computational transparency and reproducible results, we report training and inference latencies in Table 13, and the detailed hardware and software environment in Table 14. We document the compute budget and environment to support reproducibility and fair comparison. We report wall-clock time, estimated GPU hours, and per–rebalance as well as per-asset inference time by dataset, and disclose the hardware and software stacks used for all runs.

Table 13: The compute budget by each dataset for one full training run with 200 epochs using four GPUs (4×RTX 3080). Estimated GPU hours assume full utilization. Latency (rebalance) represents the end-to-end per-rebalance inference time on the full universe (55 assets for US/China, 37 for Europe, and 44 for South Korea). Latency (asset) denotes the corresponding per-asset time.

| Dataset | Wall time | Est. GPU hours | Latency (rebalance) | Latency (asset) |
|---|---|---|---|---|
| S&P500 (US) | 2h 01m 13s | 8.08 | 31.21 s | 568 ms |
| CSI300 (China) | 1h 59m 42s | 7.98 | 29.64 s | 539 ms |
| EUROSTOXX (Europe) | 1h 21m 05s | 5.41 | 21.37 s | 578 ms |
| KOSPI200 (South Korea) | 1h 35m 28s | 6.36 | 25.84 s | 587 ms |

Table 14: **Hardware and software environment.**

| Item | Specification |
|---|---|
| GPUs | 4× NVIDIA RTX 3080 (10 GiB each) |
| CPU | 2× Intel Xeon Silver 4214 @ 2.20 GHz (12 cores/socket, 24 cores, 48 threads total) |
| RAM | 503 GiB |
| Storage | 33 TiB root volume (PERC H730P RAID) |
| OS | Ubuntu 20.04.5 LTS, Linux kernel 5.4.0-216-generic |
| Driver and CUDA | NVIDIA driver 535.183.01, CUDA runtime 12.2; CUDA Toolkit 11.6 (nvcc 11.6.124) |
| Deep learning framework | PyTorch 1.13.1 (`cu116`, cuDNN 8.3.2); `torchvision` 0.14.1; `torchaudio` 0.13.1 |