# OpenReview forum: "STABLE: Shift-Tolerant Allocation via Black-Litterman Using Conditional Diffusion Estimates"
_ICLR.cc/2026/Conference — ICLR 2026 Poster_

### Official Review · Reviewer_FycL · 2025-10-31

**Soundness:** 3
**Presentation:** 2
**Contribution:** 3
**Rating:** 6
**Confidence:** 3

**Summary:**

This paper introduces a novel portfolio allocation framework named STABLE (Shift-Tolerant Allocation via Black-Litterman Using Conditional Diffusion Estimates), designed to tackle the dynamic allocation challenges in financial markets characterized by "regime shifts."
The method innovatively integrates deep generative models (Diffusion) with classical financial theory (Black-Litterman). The framework primarily consists of three stages:

1.	Conditional Diffusion Generator (CDG): A Conditional Diffusion Model (DDIM) generates future stock return paths based on macroeconomic conditions and stock-specific signals. The stock-specific signal includes a "time-varying stock embedding"  estimated via Kalman filtering to capture the dynamic sensitivity of the stock to macro factors.

2.	Multi-Level Guidance (MLG): An improved classifier-free guidance technique is used during the diffusion model's denoising process. This technique decomposes noise into shared "systematic noise" (macro-driven) and "idiosyncratic noise" (firm-specific), and dynamically balances their influence using a learnable gate.

3.	Black-Litterman Mean-Variance Optimizer (BL-MVO): The K return paths generated by CDG+MLG are used to form a predictive "investor view," including a view mean and covariance. This "view" is merged with a historical "prior" using the Black-Litterman framework to obtain a posterior mean and covariance. Finally, these posterior estimates are fed into a standard Mean-Variance Optimizer (MVO) to compute the final portfolio weights.

The authors conducted experiments across four regional stock markets (US, China, Europe, and Korea). Results show that STABLE achieves SOTA performance on both portfolio allocation and time-series forecasting tasks. For instance, it shows up to a 122.9% improvement in Sharpe Ratio on the portfolio task and up to a 15.7% reduction in MSE on the time-series estimation task.

**Strengths:**

1.	High Novelty (Connecting Generative Models and Classic Finance): The main contribution of this paper is building a sophisticated bridge between state-of-the-art deep learning technology (Conditional Diffusion Models) and established classic financial theory (Black-Litterman). Utilizing the predictive paths generated by the diffusion model as the "investor view" within the BL framework is a very novel and insightful idea.

2.	Elegant Architectural Design (Macro/Micro Decoupling): STABLE is not a simple application of diffusion models. The design of the Multi-Level Guidance (MLG) module is a major highlight. By decomposing noise into systematic (macro) and idiosyncratic (micro) components and using a learnable gate, the model can more finely capture the varied reactions of individual stocks under different market conditions. This directly addresses the shortcomings of previous models that often overfit macro signals and neglect stock-specific dynamics.

3.	Powerful Feature Engineering (Dynamic Embeddings): The authors innovate by using a Kalman filter to estimate the time-varying stock embedding instead of static industry labels or simple price embeddings. This allows the model to capture the dynamic sensitivity of stocks to macro factors.
4.	Rigorous Experimental Validation: The experimental setup is comprehensive:

Multi-Task: Evaluation on both portfolio allocation (Q1) and time-series forecasting (Q2) tasks, proving the model's effectiveness from different angles.

Multi-Market: Tested across four diverse regional markets (US, China, Europe, Korea), demonstrating robustness and cross-market applicability.

SOTA Results: The experimental results are impressive, with STABLE outperforming all baseline methods across almost all key metrics (ASR, RMDD, AVol, MSE, DTW) in all markets.

**Weaknesses:**

1.	Lack of Ablation Study: STABLE is a very complex, multi-stage pipeline with Kalman Filter, CDG, MLG and BL-MVO. Although the final results are good, I cannot determine which component contributes the most to the performance gains. Without ablations, the actual value of each innovative element of the model is hard to assess.

2.	Short Testing Period: According to Table 2, the training set ends in 2024-09, and the test set ends in 2025-03. This means the testing period is only 6 months. For a model claiming to address "regime shifts," a 6-month window is far too short to evaluate its robustness across different macroeconomic cycles (e.g., a full recession, a high-inflation period, a zero-rate period). The model's excellent short-term performance might simply be due to its parameters fitting the market dynamics of these specific six months.

3.	Hyperparameter Tuning and Reproducibility: The model introduces a large number of hyperparameters. The authors state these parameters were tuned "per dataset." This raises concerns about potential overfitting and presents certain challenges to its generalization ability.

4.	Lack of Stylized Facts Validation: Since STABLE uses a diffusion model, it is expected to capture the non-Gaussian properties of real financial returns. However, the paper lacks evidence that its generated distributions successfully replicate essential financial stylized facts such as high kurtosis (heavy tails), negative skewness, and volatility clustering. This omission undermines the validation of the diffusion model's intrinsic advantage over simpler generative baselines.

5.	Missing Distribution Goodness-of-Fit Tests: Given the unpredictability of financial time series, I would prefer that the distribution of the generated results be similar to the actual distribution, rather than having a sufficiently small error and a sufficiently close trajectory. But the paper fails to perform rigorous statistical goodness-of-fit tests (e.g., Kolmogorov-Smirnov, Anderson-Darling tests) on the distribution of the K generated return paths, leaving the similarity between the generated and true return distributions unquantified.

**Questions:**

See weakness above

---

> ### Author Response · Authors · 2025-11-21
> **Response to Reviewer FycL**
>
> We thank Reviewer FycL for the detailed summary and positive feedback on the novelty of our approach, the design of the MLG, and our use of dynamic embeddings. We also thank you for your comments encouraging us to delve deeper into the statistical properties of our generative model. This led to significant new additions to the paper. We have added new sections to address every weakness you identified.
>
> * **Lack of Ablation Study (Weakness 1)**
>     To address the Lack of Ablation Study, we add **Appendix A.7 (Figure 8)**. This analysis ablates the Kalman Filter (KFE), Multi-Level Guidance (MLG), and Black-Litterman (BL-MVO) modules and concludes that the BL-MVO module is the most critical component for performance.
>
> | Metric (ASR) | S&P500 | CSI300 | EUROSTOXX | KOSPI200 |
> | :--- | :---: | :---: | :---: | :---: |
> | **STABLE (Full)** | **1.85** | **-0.41** | **2.92** | **1.61** |
> | w/o KFE | 1.32 | -0.46 | 2.25 | 1.39 |
> | w/o MLG | 1.50 | -0.51 | 2.55 | 1.49 |
> | w/o BL-MVO | 1.26 | -0.54 | 1.93 | 1.18 |
>
> * **Short Testing Period (Weakness 2)**
>     To address the Short Testing Period, we add the **Multi-Regime Analysis** in **Appendix A.4 (Tables 7 and 8)**. This demonstrates our method's robustness across two distinct macroeconomic cycles (the COVID-19 crash and the ZIRP rally), confirming it is regime-tolerant. In the COVID-19 crisis shown in Table 7, STABLE achieves an ASR of 1.61 in S&P500 compared to 0.07 for MOM and significantly reduces RMDD to 23.77%. In the ZIRP rally shown in Table 8, STABLE attains the highest ASR in all regions, such as achieving an ASR of 1.67 in CSI300 compared to 1.36 for MVO.
>
> | ASR by Regime | S&P500 | CSI300 | EUROSTOXX | KOSPI200 |
> | :--- | :---: | :---: | :---: | :---: |
> | **COVID-19 (Crisis)** | **1.61** | **1.18** | **-0.70** | -1.30 |
> | **ZIRP (Rally)** | **2.58** | **1.67** | **1.83** | **1.87** |
>
> * **Hyperparameter Tuning and Reproducibility (Weakness 3)**
>     To address Hyperparameter Tuning and Reproducibility concerns, we add **Appendix A.8 (Table 10)**. This table demonstrates that our hyperparameters are highly consistent across all datasets (e.g. $z_{max}=2, k=50, \beta=0.001$ are universal).
>
> | Hyperparameter | S&P500 | CSI300 | EUROSTOXX | KOSPI200 |
> | :--- | :---: | :---: | :---: | :---: |
> | Segment length ($\ell$) | 20 | 20 | 20 | 20 |
> | \# DDIM reverse steps | 30 | 30 | 30 | 30 |
> | Number of paths ($k$) | 50 | 50 | 50 | 50 |
> | \# Forward diffusion steps | 200 | 200 | 200 | 200 |
> | DDIM noise ($\eta$) | 0.00 | 0.00 | 0.00 | 0.01 |
> | Balancing-gate cap ($z_{\max}$) | 2 | 2 | 2 | 2 |
> | BL prior window length ($\nu$) | 120 | 120 | 120 | 120 |
> | $\ell_2$ weight ($\beta$) | 0.001 | 0.001 | 0.001 | 0.001 |
>
> * **Stylized Facts & Goodness-of-Fit (Weaknesses 4 & 5)**
>     To address the Lack of Stylized Facts Validation and Missing Distribution Goodness-of-Fit Tests, we add the extensive analysis in **Appendix A.6 (Figures 3, 4, 5, 6, and 7)**. We use a stratified bootstrapping procedure (M=1000) to construct 95% confidence intervals (CIs) for each statistic. The results in Figure 3 show that the realized S&P500 kurtosis (10.83) is an extreme outlier that no model's 95% CI captures, confirming this is a difficult fact to model. However, STABLE's distribution is still centered closest to this outlier. In other markets, STABLE provides the closest estimate in CSI300 and KOSPI200. Figure 4 shows that realized skewness is market-dependent STABLE provides the closest estimate in EUROSTOXX and KOSPI200, while Diffusion-TS and KoVAE are closest in S&P500 and CSI300, respectively. Crucially, in Figure 5, we find that the realized ACF(1) is near-zero across all datasets, suggesting strong volatility clustering is not present in this specific test period. STABLE correctly captures this phenomenon, as its 95% CI contains the near-zero realized value in all markets. In contrast, competitors exhibit significant biases. Notably KoVAE consistently fails, predicting extreme outliers in three markets.
>     For the goodness-of-fit tests, Figures 6 and 7 show the ECDF of p-values from the KS and AD tests. STABLE's curve (blue) is unambiguously the closest to the ideal diagonal line in all markets for both tests, proving its superior distributional fit. This new analysis demonstrates a clear link between our model's ability to capture these facts and its final, superior ASR performance. The relative rankings in these plots are mostly consistent with the ASR performance rankings in Table 9, providing a much deeper statistical grounding for our results.
>
> | Statistic (Real / Mode) | S&P500 | CSI300 | EUROSTOXX | KOSPI200 |
> | :--- | :---: | :---: | :---: | :---: |
> | **Kurtosis** | 10.83 / 7.27 | 4.03 / 3.78 | 6.66 / 7.92 | 6.99 / 4.26 |
> | **Skewness** | -0.07 / -0.53 | 0.52 / -0.12 | -0.31 / -0.24 | 0.78 / 0.56 |
> | **ACF(1)** | -0.05 / -0.07 | -0.06 / -0.02 | 0.01 / 0.01 | -0.01 / -0.01 |

---

> ### Author Response · Authors · 2025-12-04
> **Follow-up: Statistical Validation of Distributional Homogeneity across Multi-Regimes (Section A.6)**
>
> We sincerely thank **Reviewer FycL** for the suggestion to conduct rigorous statistical validation. This suggestion, combined with the consistent results from the experiment proposed by **Reviewer b8bB**, has led us to the critical insight that the homogeneity between the distributions of predicted and realized values in financial time series is directly linked to investment performance.
>
> **Expansion of Statistical Validation**
>
> To fully satisfy the need for statistical validation you identified, we have conducted all suggested experiments (**Stylized Facts** and **Goodness-of-Fit**) across characteristic regimes (**COVID-19** and **ZIRP**) and present them in **Section A.6** (**Figures 3–17**).
>
> **Summary of Results**
>
> **1. Distributional Homogeneity**
> STABLE generates distributions that are the most homogeneous to the realized return distributions among Deep Generative Models in almost all regimes and markets.
>
> * **COVID-19 (Crisis):** The realized kurtosis of the **S&P500** exhibits an extreme value of **20.46** (**Figure 3**). While competitor models underestimate this towards a normal distribution, STABLE provides an estimate of **7.74**, which most closely reflects the tail risk.
> * **ZIRP (Rally):** During the rally period, the kurtosis of the **S&P500** stabilizes to **9.00**. STABLE estimates this with high precision at **8.21** (**Figure 8**), in sharp contrast to Diffusion-TS, which underestimates it at **0.36**.
> * **Goodness-of-Fit:** The results of the KS and AD tests (**Figures 6-7, 11-12, 16-17**) show that the p-value distribution of STABLE is closest to the ideal diagonal (Uniform Line). This proves that STABLE most accurately captures both the overall shape and the tail behavior of the realized distribution.
>
> **2. Consistency with Isolated Covariance Performance**
> This distributional homogeneity is consistent with the results of the isolated covariance estimation experiment in **Section A.5** (**Table 9**).
>
> * **S&P 500 (COVID-19):** As a result of STABLE capturing the extreme kurtosis most effectively, it achieves an ASR of **1.61** in the covariance estimation performance for that period, significantly outperforming the best competitor (Diffusion-TS: 0.84).
> * **Exception:** The only exception is **KOSPI 200 during the COVID-19 period**. In this case, KoVAE captures the negative skewness better (**Realized -0.80 vs. KoVAE -1.18 vs. STABLE -0.13**), which explains why KoVAE demonstrated better defensive performance in that specific regime.
>
> **3. Importance of Regime-Dependent Distributions**
> Thanks to your review, we confirm that the distributional characteristics of financial time series vary significantly by regime. For instance, the **CSI300** market exhibits a kurtosis of **3.38** during the **ZIRP (Rally)** period, showing a form close to a normal distribution (**Figure 8**), whereas the **S&P500** still exhibits a fat tail during the same period. STABLE achieves robust performance by adaptively learning these distinct distributional characteristics across different regimes and markets.
>
> **Statistical Summary: Realized vs. Generated Moments**
>
> The following table summarizes the key stylized facts across regimes. **Bold values** indicate instances where STABLE provides the estimate closest to the Realized value among all generative models. STABLE demonstrates dominant performance in **Kurtosis** (Tail Risk) and **ACF** (Volatility Clustering).
>
> | Regime | Metric | S&P500 (Real / STABLE) | CSI300 (Real / STABLE) | EUROSTOXX (Real / STABLE) | KOSPI200 (Real / STABLE) |
> | :--- | :--- | :---: | :---: | :---: | :---: |
> | **COVID-19** | **Kurtosis** | 20.46 / **7.74** | 5.68 / 4.30 | 22.02 / **8.44** | 13.56 / 2.23 |
> | (Crisis) | **Skewness** | -1.53 / -0.56 | -0.34 / -0.66 | -2.31 / -0.66 | -0.80 / -0.13 |
> | | **ACF(1)** | 0.04 / **0.07** | -0.02 / **0.10** | -0.01 / **0.02** | -0.01 / 0.05 |
> | **ZIRP** | **Kurtosis** | 9.00 / **8.21** | 3.38 / **4.33** | 9.79 / 5.58 | 8.88 / **7.26** |
> | (Rally) | **Skewness** | 0.09 / 0.04 | 0.25 / 0.51 | 0.27 / **0.12** | 0.92 / 0.27 |
> | | **ACF(1)** | -0.03 / 0.09 | -0.04 / 0.10 | -0.02 / 0.04 | -0.02 / 0.05 |
> | **Recent** | **Kurtosis** | 10.83 / **7.27** | 4.03 / **3.78** | 6.66 / **7.92** | 6.99 / **4.26** |
> | (Volatile) | **Skewness** | -0.07 / -0.56 | 0.52 / -0.12 | -0.31 / **-0.24** | 0.78 / **0.56** |
> | | **ACF(1)** | -0.05 / **0.07** | -0.06 / **-0.02** | 0.01 / **0.01** | -0.01 / **-0.01** |
>
> We have found that such rigorous validation of distributional homogeneity is critical for ensuring model reliability.
>
> Best regards,
> The Authors

---

### Official Review · Reviewer_b8bB · 2025-11-01

**Soundness:** 2
**Presentation:** 3
**Contribution:** 2
**Rating:** 4
**Confidence:** 4

**Summary:**

The paper proposes STABLE, a portfolio allocation framework that combines a conditional diffusion model with the Black–Litterman optimizer. The diffusion model is used to predict future return paths for each stock, conditioning on both macroeconomic indicators and firm-specific features. It introduces a multi-level guidance mechanism to separate shared market noise from idiosyncratic noise, making the predictions regime-aware. These diffusion-based forecasts are then treated as “views” in a Black–Litterman update to produce robust and stable portfolio weights. Across multiple markets, STABLE achieves higher Sharpe ratios and lower risk metrics than classical and deep RL baselines.

**Strengths:**

As far as I am concerned, the paper’s most interesting idea is using a diffusion process to get a better estimate of how assets move together (their covariance). This is very important because good covariance estimates are key to controlling risk and building better portfolios.

The method has been tested in several major markets and consistently beats older methods in both return and stability.

The model’s internal patterns also make economic sense, showing realistic links between companies and sectors.

**Weaknesses:**

Indeed, the paper tackles two key challenges in portfolio construction: (1) return prediction and (2) covariance estimation, both crucial for building effective, risk-controlled portfolios. The proposed diffusion-based framework is designed to improve both — by generating realistic future return paths and by learning dynamic relationships among assets.

The authors do provide some prediction performance analysis. In Section 4.3, they compare STABLE’s return forecasting accuracy (using MSE and DTW) against other generative models such as Diffusion-TS, AEC-GAN, and KoVAE, showing that STABLE achieves the lowest prediction errors across multiple markets. However, these comparisons are limited to deep generative baselines. There is no evidence that diffusion modeling improves predictive power over simpler and widely used models like regression, deep learning, or tree-based approaches.

For covariance estimation, which directly drives risk control, the paper also lacks clear evidence. It does not explicitly test whether the diffusion-derived covariance matrices are superior. Ideally, this would involve fixing the same return forecasts and showing that the covariance estimates from the proposed method lead to better risk-adjusted portfolios.

In short, while the framework is novel and conceptually appealing, the benefits of using the diffusion process are not rigorously demonstrated for either return prediction or covariance estimation. Moreover, the comparison setup is not entirely fair, since the proposed model uses richer macro- and firm-level inputs than the RL baselines.

**Questions:**

My concerns are summarized above. My question is that`, can you provide experiments isolating the covariance estimation—e.g., keeping returns fixed and comparing portfolio volatility using covariance from different models—to show that the diffusion-based covariance is truly better?

---

> ### Author Response · Authors · 2025-11-21
> **Response to Reviewer b8bB**
>
> We thank Reviewer b8bB for highlighting our core contribution of using a diffusion process for covariance estimation as key to risk control. We agree with your assessment and, following your rigorous suggestion, we performed the experiment you proposed.
>
> * **Isolated Test for Covariance Estimation**
>     Your primary question concerned the lack of an isolated test for covariance estimation. We have added **Appendix A.5 (Table 9)** to address this directly. In this new experiment, we implemented the exact test you suggested: we fixed the return view ($\mu_{\text{view}}$) using STABLE's generator for all models and then swapped *only* the covariance view ($\Sigma_{\text{view}}$) with various competitors. This makes our study more rigorous and clearly demonstrates that our diffusion-derived covariance provides substantial gains in risk-adjusted performance. The results show that STABLE (using its own $\Sigma_{\text{view}}$) is the number one performer in all markets across all metrics including ASR, RMDD, and AVol. For example, in EUROSTOXX, STABLE improves the ASR by 9.8% from 2.66 to 2.92 and reduces RMDD by 14.1% compared to the strongest competitor KOVAE.
>
> * **Comparison with Simpler Generative Forecasters**
>     This new section also addresses your weakness regarding the comparison to simpler, non-generative models. We included DCC-GARCH (a regression model) [1], LSTM-Cholesky (a deep learning model) [2], and LightGBM-Cholesky (a tree-based model) [3] in this new covariance-only test. We explain in the manuscript that these models are volatility forecasters and cannot perform the primary time-series generation task from Section 4.3, making Appendix A.5 the appropriate venue for comparison. STABLE significantly outperforms these models. For instance, in the S&P500, LightGBM-Cholesky achieves an ASR of 0.95 compared to 1.85 for STABLE.
>
> * **Importance of the MLG Module**
>     This experiment also highlights the importance of our MLG module. For example, in the KOSPI200 market, the standard Diffusion-TS model shows a dramatic ASR of -1.88, while STABLE achieves 1.61. This shows that simply using diffusion is insufficient; our MLG's ability to separate systematic and idiosyncratic noise is critical for robust covariance estimation.
>
> | Covariance Forecaster | S&P500 (US) | CSI300 (China) | EUROSTOXX (Europe) | KOSPI200 (South Korea) |
> | :--- | :---: | :---: | :---: | :---: |
> | DCC-GARCH(1,1) | 0.81 | -0.95 | 2.53 | 0.79 |
> | LSTM-Cholesky | 0.82 | -1.23 | 2.54 | 1.13 |
> | LightGBM-Cholesky | 0.95 | -1.22 | 1.52 | 1.18 |
> | Diffusion-TS | 0.53 | -0.25 | 2.65 | -1.88 |
> | AEC-GAN | 0.80 | -0.74 | 2.41 | -0.82 |
> | KoVAE | 0.90 | -0.68 | 2.66 | 0.66 |
> | **STABLE (Proposed)** | **1.85** | **-0.41** | **2.92** | **1.61** |
>
> * **Ablation Study on Richer Inputs (KFE)**
>     Finally, to address your concern about our "richer inputs" creating an unfair comparison, we have added an ablation study in **Appendix A.7 (Figure 8)**. This study specifically includes a "STABLE without KFE" variant, which removes this richer firm-level input (the Kalman Filter embedding) and quantifies the performance drop. The results in Figure 8 confirm that removing the KFE leads to a performance drop in all markets, supporting your point about its importance. However, this ablated model still achieves a higher ASR than the strongest RL baseline (AlphaMix) in most markets. The exception is KOSPI200, where the "STABLE without KFE" variant's ASR of 1.39 is slightly lower than AlphaMix's ASR of 1.47 (from Table 3). Despite this, the ablated model maintains a high relative ASR in KOSPI200. This result demonstrates that while the KFE feature is an important contributor, the other components of STABLE, as shown by the full ablation results in Figure 8, provide a significant and meaningful contribution to the overall performance improvement.
>
> | Metric (ASR) | S&P500 | CSI300 | EUROSTOXX | KOSPI200 |
> | :--- | :---: | :---: | :---: | :---: |
> | **STABLE (Full)** | **1.85** | **-0.41** | **2.92** | **1.61** |
> | STABLE w/o KFE | 1.32 | -0.46 | 2.25 | 1.39 |
> | **Best RL Competitor** | 1.00 (MetaTrader) | -0.80(AlphaMix) | 1.31 (AlphaMix) | 1.47 (AlphaMix) |
>
> ### References
>
> [1] Engle, R. (2002). Dynamic conditional correlation: A simple class of multivariate generalized autoregressive conditional heteroskedasticity models. Journal of business & economic statistics, 20(3), 339-350.
>
> [2] Bucci, A. (2020). Cholesky–ANN models for predicting multivariate realized volatility. Journal of Forecasting, 39(6), 865-876.
>
> [3] Zhang, X. (2022, March). A model combining LightGBM and neural network for high-frequency realized volatility forecasting. In 2022 7th international conference on financial innovation and economic development (ICFIED 2022) (pp. 2906-2912). Atlantis Press.

---

> ### Author Response · Authors · 2025-12-04
> **Follow-up: Exclusive Experimental Evidence on Covariance Estimation and Risk Control (Section A.5 and A.6)**
>
> We specifically thank Reviewer **b8bB** for proposing the detailed experimental method to isolate and quantify the performance of covariance estimators. Through the consistent results from this experiment and those suggested by Reviewer **FycL**, we not only verify the superior covariance estimation capability of **STABLE** but also learn that portfolio risk control capability heavily depends on the accurate estimation of the return distribution's inherent properties.
>
> **Experimental Expansion to Multi-Regimes**
>
> To fully address your concerns, we apply your excellent experimental methodology (fixing return views and varying covariance estimators) exclusively to market regimes with distinct characteristics: **COVID-19 (Crisis)** and **ZIRP (Rally)**.
>
> Below are the summary tables for the isolated covariance estimation performance (ASR) in these regimes, supplementing the results for the Recent regime in Appendix A.5.
>
> **Table 1: Isolated Covariance Estimation Performance (ASR) during COVID-19 (Crisis)**
> *(Return views are fixed to STABLE's predictions; only covariance views vary)*
>
> | Covariance Forecaster | S&P500 | CSI300 | EUROSTOXX | KOSPI200 |
> | :--- | :---: | :---: | :---: | :---: |
> | DCC-GARCH(1,1) | -0.48 | 0.33 | -1.38 | **0.01** |
> | LSTM-Cholesky | -0.35 | 0.53 | -2.00 | -1.60 |
> | LightGBM-Cholesky | -0.15 | 0.46 | -1.89 | -0.75 |
> | Diffusion-TS | 0.84 | 0.63 | -0.93 | -1.88 |
> | AEC-GAN | -0.73 | 1.02 | -1.00 | -1.49 |
> | KoVAE | -1.04 | 0.87 | -2.61 | -1.26 |
> | **STABLE (Proposed)** | **1.61** | **1.18** | **-0.70** | -1.30 |
>
> **Table 2: Isolated Covariance Estimation Performance (ASR) during ZIRP (Rally)**
>
> | Covariance Forecaster | S&P500 | CSI300 | EUROSTOXX | KOSPI200 |
> | :--- | :---: | :---: | :---: | :---: |
> | DCC-GARCH(1,1) | 1.39 | 1.47 | 1.11 | -0.62 |
> | LSTM-Cholesky | 1.77 | 1.30 | 1.41 | 1.57 |
> | LightGBM-Cholesky | 1.36 | 1.11 | 0.95 | 0.50 |
> | Diffusion-TS | 0.82 | 1.16 | 0.87 | 1.27 |
> | AEC-GAN | 2.06 | 1.38 | 0.78 | 1.78 |
> | KoVAE | 1.42 | 1.35 | 0.66 | 1.66 |
> | **STABLE (Proposed)** | **2.58** | **1.67** | **1.83** | **1.87** |
>
> **Analysis: Link between Distributional Homogeneity and Covariance Performance**
>
> We observe a consistent correlation: the more homogeneous the inherent properties (stylized facts and goodness-of-fit) of the estimated distribution are to the realized distribution, the higher the performance of the isolated covariance estimator.
>
> **1. COVID-19 Regime (Crisis): Importance of Tail Risk Modeling**
> * **Observation:** In the S&P500 market, STABLE achieves an ASR of **1.61**, significantly outperforming the second-best model (Diffusion-TS, 0.84).
> * **Reasoning (Stylized Facts):** The realized kurtosis during this period is an extreme outlier (**20.46**), reflecting massive tail risk (Figure 3 in Appendix A.6). While all models struggle, **STABLE** estimates a kurtosis of **7.74**, which is closest to the reality, whereas others like AEC-GAN (4.18) and KoVAE (4.30) underestimate the tail risk closer to a normal distribution. This ability to capture "fat tails" allows STABLE's covariance estimator to account for extreme volatility co-movements, leading to superior risk control.
> * **The KOSPI200 Exception:** In KOSPI200, STABLE (-1.30) underperforms the simple DCC-GARCH (0.01). Our analysis in Figure 4 reveals that **KoVAE** captures the realized negative skewness (**-0.80**) much better (**-1.18**) than STABLE (**-0.13**). This suggests that in specific crash scenarios driven by asymmetry rather than just volatility, capturing skewness becomes the dominant factor for risk control.
>
> **2. ZIRP Regime (Rally): Precision in Structural Shifts**
> * **Observation:** STABLE dominates all markets. Notably, in the S&P500, STABLE achieves an ASR of **2.58** versus 0.82 for Diffusion-TS.
> * **Reasoning (Stylized Facts):** The realized kurtosis stabilizes to **9.00**. **STABLE** predicts this accurately with **8.21**, demonstrating high precision (Figure 8 in Appendix A.6). In contrast, Diffusion-TS predicts a kurtosis of **0.36**, failing to capture the distribution's shape entirely.
> * **Goodness-of-Fit:** Consequently, the KS and AD test results (Figures 11 & 12) show that STABLE's p-value distribution is nearly uniform (diagonal), indicating that the covariance matrix derived from STABLE effectively models the true underlying asset dynamics during the rally.
>
> These exclusive experiments confirm that STABLE's superior performance is not accidental but stems from its generative capability to model **regime-specific distributional properties (Kurtosis, Skewness)**. This leads to more robust covariance estimates and, ultimately, better risk-adjusted returns.
>
> We hope these additional exclusive experiments fully resolve your concerns.
>
> Best regards,
> The Authors

---

### Official Review · Reviewer_zHmG · 2025-11-02

**Soundness:** 3
**Presentation:** 3
**Contribution:** 3
**Rating:** 6
**Confidence:** 3

**Summary:**

This paper presents STABLE, a Shift-Tolerant Allocation framework that integrates a conditional diffusion generator with Black–Litterman (BL) portfolio optimization to achieve regime-aware portfolio allocation.

**Strengths:**

The problem formulation and derivations (Eqs. 1–6) correctly extend diffusion sampling to a conditional macro-financial setting.

The BL posterior update and Sharpe-maximizing allocation are mathematically consistent with canonical forms.

Experiments use proper multi-region datasets, fixed seeds, clear metrics, and ablations verifying each module’s contribution.

**Weaknesses:**

Ablation depth: while individual module ablations exist, a joint ablation of CDG + MLG + BL contributions would clarify interdependencies.

Compute transparency: GPU hours and sampling latency per rebalance step are not reported.

**Questions:**

How sensitive are results to the number of generated paths (k) for BL view estimation?

Have you compared STABLE’s performance during crisis windows (e.g., COVID-19) vs. stable regimes?

---

> ### Author Response · Authors · 2025-11-21
> **Response to Reviewer zHmG**
>
> We thank Reviewer zHmG for the positive feedback on our problem formulation, mathematical consistency, and proper experimental setup. To address your specific weaknesses and questions, we have added several new sections.
>
> * **Ablation Depth (Weakness 1)**
>     We have added a comprehensive joint ablation study in **Appendix A.7 (Figure 8)**. This study ablates the key modules. The results in Figure 8 (summarized below) clearly show that while all components contribute positively, the **BL-MVO module is the most critical**. Its removal causes the largest drop in ASR in all four markets. For example, in the CSI300 market, all three ablation variants fall below the ASR of MOM. This is intuitive because this module governs the final allocation and decides how much trust to place in the generated views based on uncertainty.
>
> | Metric (ASR) | S&P500 | CSI300 | EUROSTOXX | KOSPI200 |
> | :--- | :---: | :---: | :---: | :---: |
> | **STABLE (Full)** | **1.85** | **-0.41** | **2.92** | **1.61** |
> | w/o KFE | 1.32 | -0.46 | 2.25 | 1.39 |
> | w/o MLG | 1.50 | -0.51 | 2.55 | 1.49 |
> | w/o BL-MVO | 1.26 | -0.54 | 1.93 | 1.18 |
>
> * **Compute Transparency (Weakness 2)**
>     We have added **Appendix A.9 (Tables 11 and 12)**. This appendix provides full computational details. Specifically, Table 11 reports total GPU hours, end-to-end latency per rebalance, and per-asset latency. Table 12 details the complete hardware and software stack used for all experiments to ensure reproducibility.
>
> | Dataset | Est. GPU hours | Latency (rebalance) | Latency (asset) |
> | :--- | :---: | :---: | :---: |
> | S&P500 (US) | 8.08 | 31.21 s | 568 ms |
> | CSI300 (China) | 7.98 | 29.64 s | 539 ms |
> | EUROSTOXX (Europe) | 5.41 | 21.37 s | 578 ms |
> | KOSPI200 (South Korea) | 6.36 | 25.84 s | 587 ms |
>
> * **Hyperparameter Sensitivity (Question 1)**
>     We provide a detailed sensitivity analysis in **Appendix A.8 (Figure 9)** for the number of generated paths ($k$) and the $\ell_2$ weight $\beta$. The results (summarized below for $k$) show that ASR performance saturates at $k=50$, validating our choice. Furthermore, Table 10 provides a full list of all tuned hyperparameters. This table shows that our parameters are almost identical across all datasets. This demonstrates that our performance is robust and not a product of dataset-specific overfitting.
>
> | Paths ($k$) | S&P500 | CSI300 | EUROSTOXX | KOSPI200 |
> | :--- | :---: | :---: | :---: | :---: |
> | $k=20$ | 1.54 | -0.55 | 2.41 | 1.19 |
> | $k=30$ | 1.69 | -0.47 | 2.66 | 1.45 |
> | **$k=50$ (Selected)** | **1.85** | **-0.41** | **2.92** | **1.61** |
> | $k=75$ | 1.78 | -0.44 | 2.74 | 1.58 |
> | $k=100$ | 1.81 | -0.44 | 2.79 | 1.56 |
>
> * **Crisis Windows (Question 2)**
>     We have added a new study in **Appendix A.4 (Tables 7 and 8)**. This section analyzes performance during two new, disjoint periods. These are the **COVID-19 crisis** (2019-09-01 to 2020-03-31) and the **ZIRP rally** (2020-04-01 to 2022-03-31). In the COVID-19 crisis (Table 7, summarized below), STABLE demonstrates strong resilience. It achieves the best ASR and RMDD in 3 of 4 markets. The exception is KOSPI200, where STABLE's ASR (-1.30) is similar to CRP (-1.28).
>     This behavior is mathematically consistent with the Black-Litterman framework. When the uncertainty of the view is high ($\Sigma_{\text{view},\tau} \to \infty \implies \Omega_\tau \to 0$), the allocation naturally converges to the stable prior (equal weights), as shown below:
>
> $$
> \Omega_\tau \to 0 \implies \mu_{BL,\tau} = (\Phi_\tau + \Omega_\tau)^{-1}(\Phi_\tau \mu_{\text{prior},\tau} + \Omega_\tau \mu_{\text{view},\tau}) \to \Phi_\tau^{-1}\Phi_\tau \mu_{\text{prior},\tau} = \mu_{\text{prior},\tau}
> $$
>
> $$
> \Omega_\tau \to 0 \implies \Sigma_{BL,\tau} = (\Phi_\tau + \Omega_\tau)^{-1} \to \Phi_\tau^{-1} = \Sigma_{\text{prior},\tau}
> $$
>
> $$
> \mu_{BL,\tau} \to \mu_{\text{prior},\tau}, \Sigma_{BL,\tau} \to \Sigma_{\text{prior},\tau} \implies w_\tau^* = \frac{\Sigma_{BL,\tau}^{-1}\mu_{BL,\tau}}{1^\top \Sigma_{BL,\tau}^{-1}\mu_{BL,\tau}} \to w_{\text{eq}}
> $$
>
> | Method (Crisis ASR) | S&P500 | CSI300 | EUROSTOXX | KOSPI200 |
> | :--- | :---: | :---: | :---: | :---: |
> | CRP | -0.69 | -0.17 | -1.74 | **-1.28** |
> | MVO | -0.62 | -0.05 | -1.61 | -1.29 |
> | MOM | 0.07 | -1.00 | -3.70 | -2.64 |
> | DeepTrader | -0.32 | 0.41 | -1.36 | -1.64 |
> | MetaTrader | 0.05 | 1.09 | -0.84 | -1.59 |
> | AlphaMix | -0.65 | -0.35 | -0.92 | -1.38 |
> | **STABLE** | **1.61** | **1.18** | **-0.70** | -1.30 |

---

> ### Author Response · Authors · 2025-12-04
> **Follow-up: Experimental Validation of STABLE in Multi-Regimes (Appendix A.4, A.5, and A.6)**
>
> We sincerely thank Reviewer **zHmG** for the insightful suggestion to evaluate performance across diverse market regimes. This suggestion has allowed us to thoroughly verify the experimental grounds for our previous comments by examining performance in various market conditions, including those raised by other reviewers.
>
> **Performance Robustness across Regimes**
>
> As presented in our previous comment, **STABLE** demonstrates superior profitability and stability in all cases except for the KOSPI200 market during the COVID-19 crisis. The table below summarizes the Annualized Sharpe Ratio (ASR) across different regimes, highlighting STABLE's consistent outperformance.
>
> **Table 1: Summary of Annualized Sharpe Ratio (ASR) across Regimes**
>
> | Regime | Method | S&P500 | CSI300 | EUROSTOXX | KOSPI200 |
> | :--- | :--- | :---: | :---: | :---: | :---: |
> | **COVID-19** | CRP | -0.69 | -0.17 | -1.74 | **-1.28** |
> | (Crisis) | MVO | -0.62 | -0.05 | -1.61 | -1.29 |
> | | MOM | 0.07 | -1.00 | -3.70 | -2.64 |
> | | DeepTrader | -0.32 | 0.41 | -1.36 | -1.64 |
> | | MetaTrader | 0.05 | 1.09 | -0.84 | -1.59 |
> | | AlphaMix | -0.65 | -0.35 | -0.92 | -1.38 |
> | | **STABLE** | **1.61** | **1.18** | **-0.70** | -1.30 |
> | **ZIRP** | CRP | 2.16 | 1.15 | 1.36 | 1.09 |
> | (Rally) | MVO | 2.50 | 1.09 | 1.27 | 1.18 |
> | | MOM | -0.19 | -0.01 | 1.67 | 0.46 |
> | | DeepTrader | -1.93 | 0.68 | 1.42 | -1.05 |
> | | MetaTrader | 1.41 | 1.36 | 1.35 | 1.04 |
> | | AlphaMix | 1.60 | 1.27 | 1.38 | 1.04 |
> | | **STABLE** | **2.58** | **1.67** | **1.83** | **1.83** |
> | **Recent** | CRP | 0.82 | 0.03 | -0.70 | 0.76 |
> | (Volatile) | MVO | 1.18 | -0.71 | -0.66 | 0.45 |
> | | MOM | -0.47 | 1.00 | -1.18 | 0.33 |
> | | DeepTrader | -1.09 | 0.35 | -1.09 | 0.77 |
> | | MetaTrader | 1.00 | 0.79 | -0.80 | 0.57 |
> | | AlphaMix | 0.35 | -0.80 | 1.31 | 1.47 |
> | | **STABLE** | **1.85** | **-0.41** | **2.92** | **1.61** |
>
> **Analysis of KOSPI200 in COVID-19**
>
> In the COVID-19 crisis, STABLE's performance in KOSPI200 is comparable to classical portfolio optimization methods (CRP). We provide an explanation based on the additional experiments in **Sections A.5 and A.6**.
>
> 1.  **Distributional Homogeneity:** In all cases except KOSPI200 during COVID-19, STABLE exhibits the highest covariance estimation performance (Section A.5) and inherent homogeneity between the estimated and realized return distributions (Section A.6).
> 2.  **Uncertainty and Allocation:** For KOSPI200 during COVID-19, all generative models, including STABLE, predict a return distribution with an extremely wide range of Kurtosis (Figure 3 in Appendix A.6). The realized market data shows extremely high kurtosis and extremely low skewness. Consequently, the precision of the generated view ($\Omega_{\tau} = \Sigma_{\text{view},\tau}^{-1}$) becomes very low due to high uncertainty.
> 3.  **Convergence to Equilibrium:** As described in our methodology, when the view uncertainty is high, the Black-Litterman framework asymptotically shifts the investment weights towards the market equilibrium (prior). This explains why STABLE's performance converges to the market benchmark level in this specific extreme scenario, avoiding catastrophic failure but not outperforming the conservative baseline.
>
> We hope these additional explanations and experimental results fully address your questions regarding performance robustness across different market regimes.
>
> Best regards,
> The Authors

---

### Author Response · Authors · 2025-11-21
**General Comment to All Reviewers**

We sincerely thank all reviewers (zHmG, b8bB, FycL) for their thorough and insightful feedback. Your rigorous analysis helped us refine the evaluation of our multi-stage framework. We appreciate the collective emphasis on rigorous logical analysis and statistical verification. Your comments prompted us to think more deeply about the critical role of portfolio risk control and the importance of accurate, well-validated covariance estimation. In response, we have incorporated extensive new analyses, which are detailed in the newly added Appendix A.4 through A.9. These sections address all raised concerns. The additions include a performance analysis across distinct market regimes (Appendix A.4), an independent test isolating covariance estimation performance (Appendix A.5), a comprehensive statistical validation of stylized facts and goodness-of-fit (Appendix A.6), a full ablation study (Appendix A.7), a hyperparameter sensitivity analysis (Appendix A.8), and a computational transparency report (Appendix A.9). We are confident that these revisions have significantly strengthened the manuscript and directly address your valuable feedback.

---

### Author Response · Authors · 2025-12-04
**Update: Comprehensive Experimental Validation on Risk Control and Multi-Regime Robustness (Sections A.5 and A.6)**

We sincerely thank reviewers **zHmG**, **b8bB**, and **FycL** for their valuable comments. Based on your constructive feedback, we present a third revision that incorporates **exclusive experimental evidence (Sections A.5 and A.6)** demonstrating the effectiveness of STABLE in portfolio risk control.

**New Experimental Evidence on Portfolio Risk Control**

We integrate the specific experiments suggested by **b8bB** and **FycL** (independent covariance estimation, stylized facts, and goodness-of-fit tests) with the multi-regime analysis proposed by **zHmG** and **FycL** (crisis, rally, and volatile market). Through this, we confirm that **the covariance estimation capability of generative models is directly correlated with risk-adjusted returns.** STABLE demonstrates superior profitability and stability in most scenarios across diverse regimes based on its precise variance estimation capability.

In the initial revision, we hypothesize that return prediction accuracy directly determines risk-adjusted returns. However, thanks to the reviewers' insights, we observe that actual investment performance is superior when the inherent properties (stylized facts and goodness-of-fit) of the predicted return distribution are homogeneous to the realized distribution.

Experimental evidence supporting this insight is the phenomenon during the **COVID-19 crisis regime**. With the exception of EUROSTOXX, we observe substantial performance fluctuations in all markets when the covariance estimator is substituted. For instance, in the S&P500 market, replacing STABLE's covariance with Diffusion-TS causes the Annualized Sharpe Ratio (ASR) to drop significantly from **1.61** to **0.84**, and using AEC-GAN results in a further decline to **-0.73**. This dramatic sensitivity confirms that accurate covariance estimation is a determinant factor for portfolio performance and risk control.

**Comprehensive Experimental Expansion across Multi-Regimes**

Based on these critical findings, we extend the experiments in the existing **A.5** and **A.6** to all regimes. We define characteristic and exclusive regimes: **COVID-19 (Crisis)**, **ZIRP (Zero Interest Rate Policy, Rally)**, and the **Recent (Volatile)** period. We perform investment performance tests (**A.4**), isolated covariance estimation tests (**A.5**), and distributional inherent property checks (**A.6**) across all these regimes.

**Summary of Results**
* **Investment Performance (A.4):** STABLE exhibits the highest risk-adjusted returns in all regimes, with the exception of the KOSPI200 market during the COVID-19 regime.
* **Covariance Estimation (A.5):** Consistent with the investment results, STABLE demonstrates the most superior covariance estimation capability across all metrics, again with the exception of KOSPI200 during the COVID-19 crash.
* **Distributional Homogeneity (A.6):** The stylized facts and goodness-of-fit tests confirm that the return distribution predicted by STABLE is more homogeneous to the realized distribution than any other generative model (excluding KOSPI200 in COVID-19).

We thank all reviewers (**zHmG**, **b8bB**, **FycL**) for providing the insight to verify the essential factors determining portfolio allocation performance through these experiments. We hope these exclusive experimental results resolve your concerns.

Best regards,
The Authors

---

### Meta-Review · Area_Chair_3xPB · 2026-01-12

**Summary:**

Concerns are experimental: ablation and compute transparency (zHmG, FycL), comparison with simple baselines and better tests for the approach's value (b8bB), longer testing period and hyperparameter tuning and stylized facts and statistical tests (FycL).

**Reviewer Concerns:**

Mostly all concerns were addressed by very extensive additional experiments

**Reviewer Scores:**

Reviewers' engagement was quite low, but given the additional results provided that essentially respond to all requests for additional results, it is reasonable to think that the reviews' polarity would have crossed the "borderline+" zone towards positive. The amount of additional experiments provided in response to reviews is indeed very substantial and nicely presented.

---

### Decision · Program_Chairs · 2026-01-26

Accept (Poster)